# Dataset of Stable Isotopes of Precipitation in the Eurasian Continent

Longhu Chen[1,2,3, a], Qinqin Wang[1,2,3, a], Guofeng Zhu[1,2,3*], Xinrui Lin[1,2,3], Dongdong Qiu[1,2,3], Yinying Jiao[1,2,3], Siyu Lu[1,2,3], Rui Li[1,2,3], Gaojia Meng[1,2,3], Yuhao Wang[1,2,3]

[1] *School of Geography and Environment Science, Northwest Normal University, Lanzhou 730070, Gansu, China*

[2] *Shiyang River Ecological Environment Observation Station, Northwest Normal University, Lanzhou 730070, Gansu, China*

[3] *Key Laboratory of Resource Environment and Sustainable Development of Oasis, Lanzhou 730000, China*

[a] *These authors contributed equally to this work and should be considered co-first authors.*

*Correspondence to: zhugf@nwnu.edu.cn*

**Abstract:** Stable isotopes in precipitation can effectively reveal the process of atmospheric water circulation, serving as an effective tool for hydrological and water resources research, climate change, and ecosystem studies. The scarcity of stable isotope data in precipitation has hindered comprehension of regional hydrology, climate, and ecology due to discontinuities on a temporal scale and unevenness on a spatial scale. To this end, we collated stable hydrogen and oxygen isotope data in precipitation from 842 stations in Eurasia from 1961 to 2022, totalling 51,752 data records. Stable isotopes in precipitation across various regions of Eurasia, as a whole, decrease with increasing latitude and distance from the coast. In the summer, stable isotopes in precipitation are relatively enriched, while in winter, they are relatively depleted. In recent decades, the stable isotope values of Eurasian precipitation show an overall trend of increasing variation with the advancement of years, which is associated with global warming. Geographical location, underlying surface conditions, seasons, and atmospheric circulation are all factors that determine the characteristics of stable isotopes in precipitation. The dataset of stable isotopes in Eurasian precipitation provides a powerful tool for understanding changes in regional atmospheric water circulation and assists in conducting hydrological, meteorological, and ecological studies in related regions.

**Keywords:** Eurasian Continent, Climate Change, Stable Isotopes in Precipitation, Atmospheric Circulation, Dataset.

## 1. Introduction

In recent years, the impacts of global climate change have become increasingly severe, particularly the significant increase in the frequency of various types of extreme weather and climate events (Faranda et al., 2023; Liu et al., 2022; Zhang et al., 2016). The World Meteorological Organization's 2022 report on the state of the climate in Asia shows that the rate of warming in Asia is higher than the global average, with droughts, floods, and heatwaves affecting most parts of the world (State of the Climate in Asia 2022). Severe fluctuations in climatic elements can alter water circulation processes, affect regional climate change, and even change the evolutionary patterns of ecological environments. Among these, stable isotopes in precipitation are an excellent comprehensive tracer, playing an important role in revealing water cycle processes, climate change information, and mechanisms of water resource use in ecosystems (Bowen et al., 2019; Wang et al., 2022). Therefore, in the face of increasingly complex climate conditions, we need more comprehensive data on stable isotopes in precipitation at various spacetime scales to help understand climate change phenomena.

Stable isotopes in precipitation serve as a crucial medium connecting the hydrological and climatic systems. Precipitation, being both a product of the climate system and a primary source for the hydrological system (Sun et al., 2018), plays a pivotal role. Additionally, stable isotope fractionation accompanying the water cycle not only carries rich climate information throughout its variations but also facilitates the tracing of contributions to various surface water bodies (Hao et al., 2019; Ren et al., 2017; Shi et al., 2022). Although stable isotopes in precipitation ($\delta^2H$ and $\delta^{18}O$) constitute a small proportion in natural water bodies, they exhibit sensitivity to changes in climatic factors (Craig, 1961; Dansgaard, 1964). The quantification of precipitation stable isotopes, influenced by factors such as temperature, precipitation, wind speed, relative humidity, and water vapour sources (Gat, 1996; Jiao et al., 2020), deepens our procedural understanding of the water cycle. This quantification provides relevant information about water vapour transport processes and precipitation formation (Kathayat et al., 2021), determination of the proportions of different types of precipitation (Aggarwal et al., 2016), and comprehension of the mechanisms behind extreme events (Sun et al., 2022), offering robust evidence to explore the inherent mechanisms of meteorological events and climate

change processes. Water recovery is a significant component of land water flux (Jasechko et al., 2013), but its direct measurement still faces numerous challenges. Deuterium excess (d-excess): $\delta^2 H = 8 \times \delta^{18}O$, a stable isotope quantity sensitive to water recovery effects, remains constant throughout the entire process from water vapor evaporation into the atmosphere to final condensation and rain formation (Merlivat and Jouzel, 1979). Therefore, in current water recovery quantification efforts, precipitation stable isotopes are a primary means (Cropper et al., 2021; Zhang et al., 2021a). $\delta^2 H$ and $\delta^{18}O$, as important climate tracers, are also employed in reconstructing continental paleoclimate. The accurate understanding of precipitation stable isotopes' response to modern climate lays the foundation for paleoclimate reconstruction. On the other hand, using general atmospheric circulation models to simulate isotope circulation is a major method for comparing isotope distributions in precipitation under both modern and ancient conditions(Joussaume et al., 1984; Brady et al., 2019). Simultaneously, the comparison between simulated and observed precipitation stable isotopes provides valuable validation for the physical components of atmospheric circulation models (Joussaume et al., 1984; Ruan et al., 2019). In conclusion, the comprehensive data on stable isotopes in precipitation offer more detailed information about the climate and hydrological systems.

In 1961, the International Atomic Energy Agency (IAEA) and the World Meteorological Organization (WMO) began establishing the Global Network for Isotopes in Precipitation (GNIP), which is the world's primary observation system. To date, research on stable isotopes in precipitation primarily relies on the GNIP database. However, GNIP's observations are very unevenly distributed in time and space. Global and regional-scale research on stable isotopes in precipitation mainly depends on model simulations. The relationship between predicted data from models and actual measured data is "comparative"(Joussaume et al., 1984). Although model simulations can compensate for the absence of measured data and are particularly advantageous in revealing the operating mechanisms of large-scale climate systems and water cycles, existing models for stable isotopes in precipitation are often insufficiently accurate. They cannot check long-term trends or characteristics of interannual variation. By integrating independent data to provide a higher density of data, it's possible to enhance the precision of model simulations.

89  We have compiled stable isotopes in precipitation data from the Eurasian continent since

90 1961 with the aim of providing more comprehensive data support for the following research

91 areas:

92  Climate research: stable isotopes in precipitation exhibit geographical and seasonal

93 variations, which can be used to study climate change and the impact of solar radiation. By

94 comparing and analyzing the stable isotopes of precipitation in different regions of the

95 Eurasian continent, long-term climate trends can be revealed, such as changes in precipitation

96 distribution and the evolution of monsoon systems.

97  Earth system research: stable isotopes in precipitation are not only influenced by climate

98 and water cycle but also by geological and biological processes. By integrating precipitation

99 stable isotope data from the Eurasian continent, it is possible to investigate in-depth the

100 interactions between different components of the Earth system, such as the interaction

101 between the atmosphere and the ocean, and the water cycle in terrestrial ecosystems. This will

102 contribute to a better understanding of the functioning and changes of the Earth system.

103  Water cycle research: stable isotopes in precipitation serve as important indicators of the

104 water cycle and can track the sources, evaporation, and precipitation processes of water. By

105 analyzing the spatial distribution and variations of precipitation stable isotopes on the

106 Eurasian continent, it is possible to understand the processes of water evaporation,

107 precipitation, and recycling, revealing the patterns of water resource distribution and changes.

108 This provides support for water resource management and hydrological modelling.

109  Paleoclimate Reconstruction: Well-established precipitation stable isotope observational

110 data are advantageous for validating paleoclimate models under modern conditions.

111 Simultaneously, they contribute to richer comparative data for stable isotopes in precipitation

112 collected in geological archives.

113 **2. Study area**

114  The Eurasian continent (10°45'N - 77°44'N, 9°30'W - 169°45'E) spans a vast territory,

115 with considerable variations in natural geographic conditions within the region (Fig.1).

116 Significant thermal differences between sea and land have given rise to a typical monsoon

117 climate system on the southeast coast, while interactions between Atlantic moisture and

118 planetary wind systems result in the west coast and wide inland areas being perennially

subject to westerly moisture. These two major systems play significant roles in global climate systems (Li et al., 2022; Wang et al., 2010). Moreover, the interactions across multiple heat zones with sea and land provide conditions conducive to a wide variety of climate types. The uplift of the Qinghai-Tibet plateau not only alters the climate patterns dominated by the planetary wind system on the Eurasian continent and the moisture movement paths in the Indian Ocean (An et al., 2001) but also changes the natural surface conditions, such as the numerous rivers including the Yangtze, Yellow, Ganges, and Mekong Rivers, which play a vital role in hydrological processes and human life. The plateau itself forms a relatively complete vertical ecological environment differentiation, enhancing the complexity of the natural environment on the Eurasian continent. Therefore, the research data and studies on climate environmental changes in Eurasia hold significant representativeness in addressing global changes.

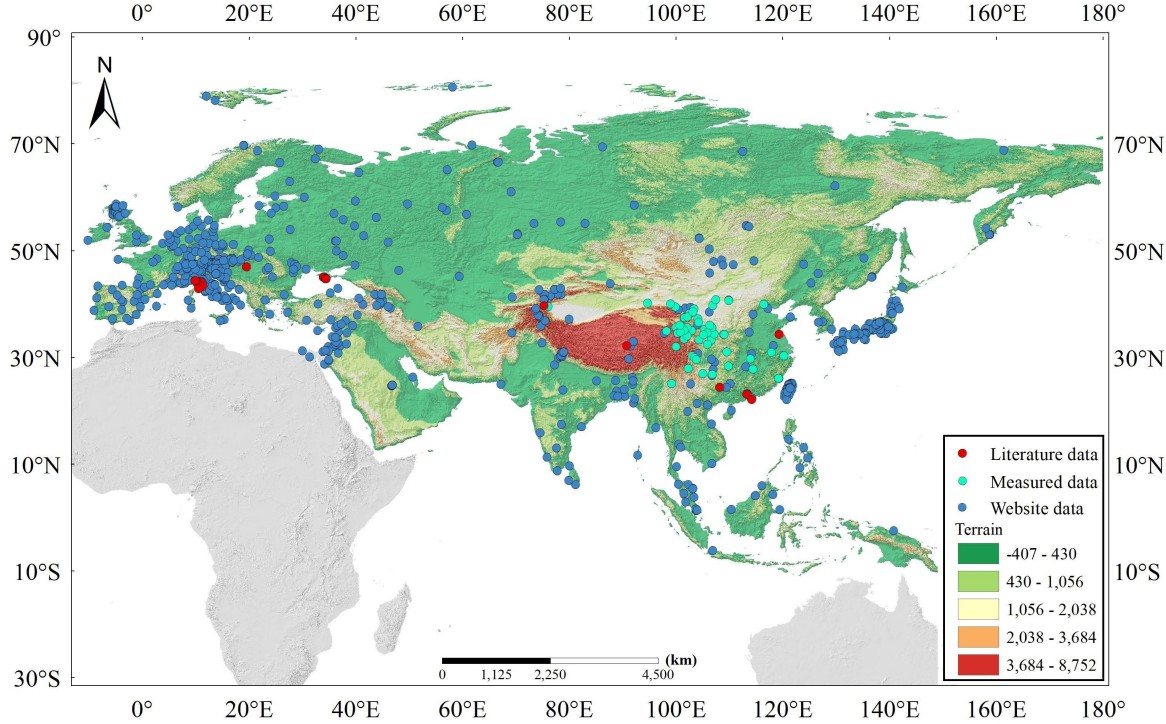

**Fig.1** Distribution map of precipitation stable isotope sampling sites in the Eurasian continent

**3. Data and methodology**

3.1 Data sources and collection

We have collected δ¹⁸O and δ²H stable isotope data from precipitation at 842 sa

mpling points across the Eurasian continent from 1961 to 2022(Supplement, Table S1).
The dataset includes both measured data and data collected from various sources. Th
e data collected are primarily from the Water Isotopes website (https://wateriso.utah.edu
/waterisotopes/index.html) and the Global Network of Stable Isotopes in Precipitation
(GNIP) operated by the International Atomic Energy Agency (IAEA). In this study, we
have compiled a total of 45,782 data records, including 3,676 records from literature
sources. The measured data were collected, analyzed, and organized at the Shiyang R
iver Basin Integrated Observation Station of Northwest Normal University in China, co
mprising 2,297 data records. Additionally, meteorological data used in this study are fr
om the CRU TS v. 4.07 dataset (Harris et al., 2020) and the NCEP-NCAR Reanalysis
1 dataset (https://psl.noaa.gov/data/gridded/data.ncep.reanalysis.html). As well as, As w
ell as, the global climate classification data of Köppen (Beck et al., 2018) (Suppleme
nt, S2).
3.2 Data processing steps and quality control
Data Collection: The data collected includes a variety of issues such as missing values,
outliers, and duplicates, as well as gaps in dates and missing or incorrect latitude and
longitude information. Therefore, the collected raw data underwent preprocessing and data
cleaning. Missing data was interpolated, entries that could not be completed were removed,
and duplicate data was eliminated.
Measured Data: Standard rain gauges were used to collect precipitation samples. After
each precipitation event, the collected samples were immediately transferred into 100ml
high-density sample bottles. To prevent data errors caused by evaporation, the collected water
samples were stored in a refrigerator at a temperature of approximately 4°C. Prior to analysis,
the precipitation samples were naturally thawed at room temperature. Impurities were filtered
out using a 0.45μm filter membrane, and the samples were transferred to 2ml sample bottles.
Isotope values were measured using a liquid water isotope analyzer (DLT-100, Los Gatos
Research, USA). For any abnormal or values that did not pass the LWIA post-analysis
software check, parallel samples were selected for re-measurement to ensure data accuracy
(Zhu et al., 2022). The isotopic abundances of $^{18}O$ and $^{2}H$ were expressed using the δ notation
relative to the International Atomic Energy Agency (IAEA) Vienna Standard Mean Ocean
Water (V-SMOW) reference, following the equation:
$$\delta_{\text{sample}}(\text{‰}) = [\frac{R_{sample}}{R_{V-smow}} - 1] \times 1000$$

Here, R represents the ratio of the heavier isotope to the lighter isotope (i.e., $^{18}O/^{16}O$ or
$^{2}H/^{1}H$). We used the International Atomic Energy Agency (IAEA) standard (V-SMOW2) to
validate our isotope measurements, ensuring comparability between isotopic measurements
across laboratories and instruments.
In 1982, Ferronsky VI and Polyakov VA conducted a study that found a general
distribution of $\delta^{18}O$ and $\delta^{2}H$ values in natural substances, indicating that the range of stable
isotope values for hydrogen and oxygen in atmospheric precipitation is typically -400‰ to
-30‰ and -60‰ to 10‰, respectively (Ferronsky VI et al., 1982). After data processing, the
data generally falls within the reasonable range.
In addition, we selected the two climatic zones with the most significant differences,
namely the tropical and polar zones. The reason for this choice is that the boundaries between
temperate, frigid, and arid zones are relatively unclear, with subtle changes in trends.
Mann-Kendall (MK) tests were conducted on the temporal variations of stable isotopes in
precipitation for both climatic zones (Fig.3). For the tropical climate (A), the stable isotopes
of precipitation ($\delta2H$ and $\delta18O$) exhibit multiple non-significant periods of abrupt changes.
There is a significant increasing trend from 1971 to 2005, followed by a non-significant
decreasing trend since 2009. Overall, the deuterium excess (d-excess) shows a non-significant
decreasing trend, but this trend has weakened after 1990. In the polar climate (E), there is a
significant increasing trend before 1973, followed by non-significant periods of both increase
and decrease after 1975. However, after 2010, a gradually significant increasing trend is
observed. Since 1985, the deuterium excess has undergone a non-significant decreasing
process, and after 2010, it gradually reaches a significant increasing trend. The uncertainty of
the tests is mainly attributed to the spatiotemporal distribution and volume of the data.

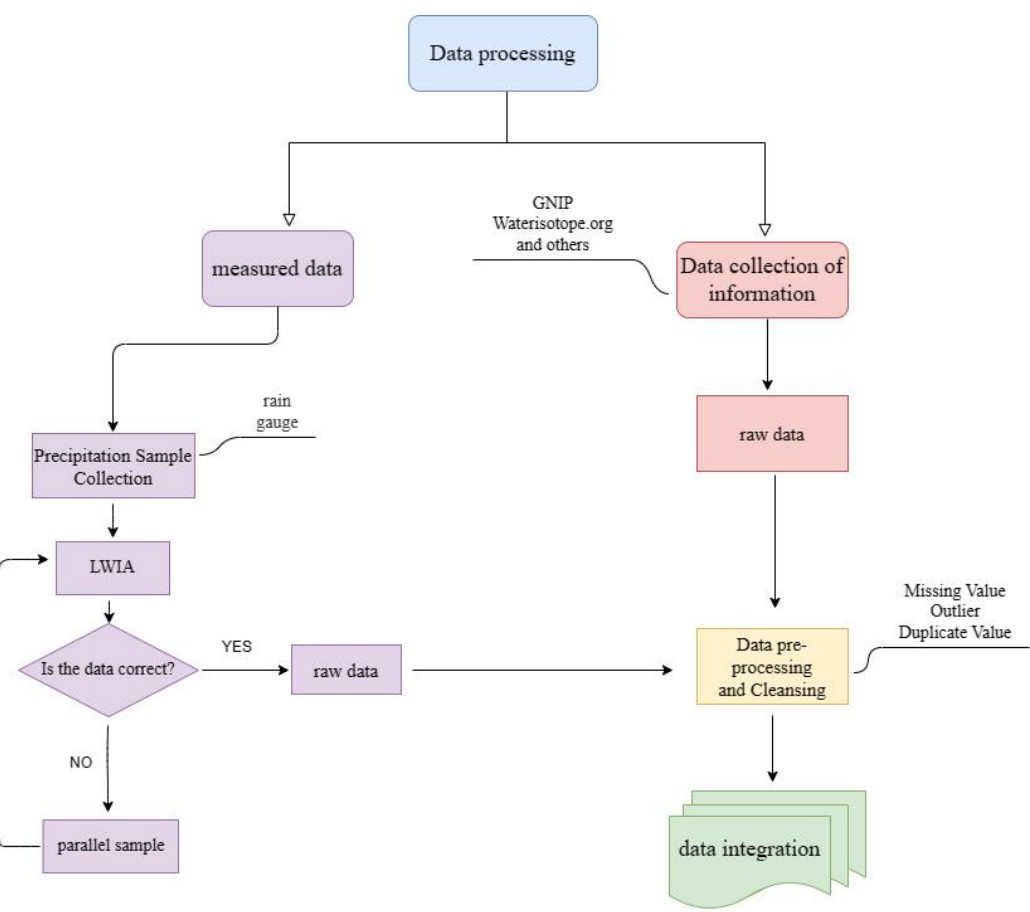

**Fig.2** Flowchart of precipitation stable isotope dataset construction

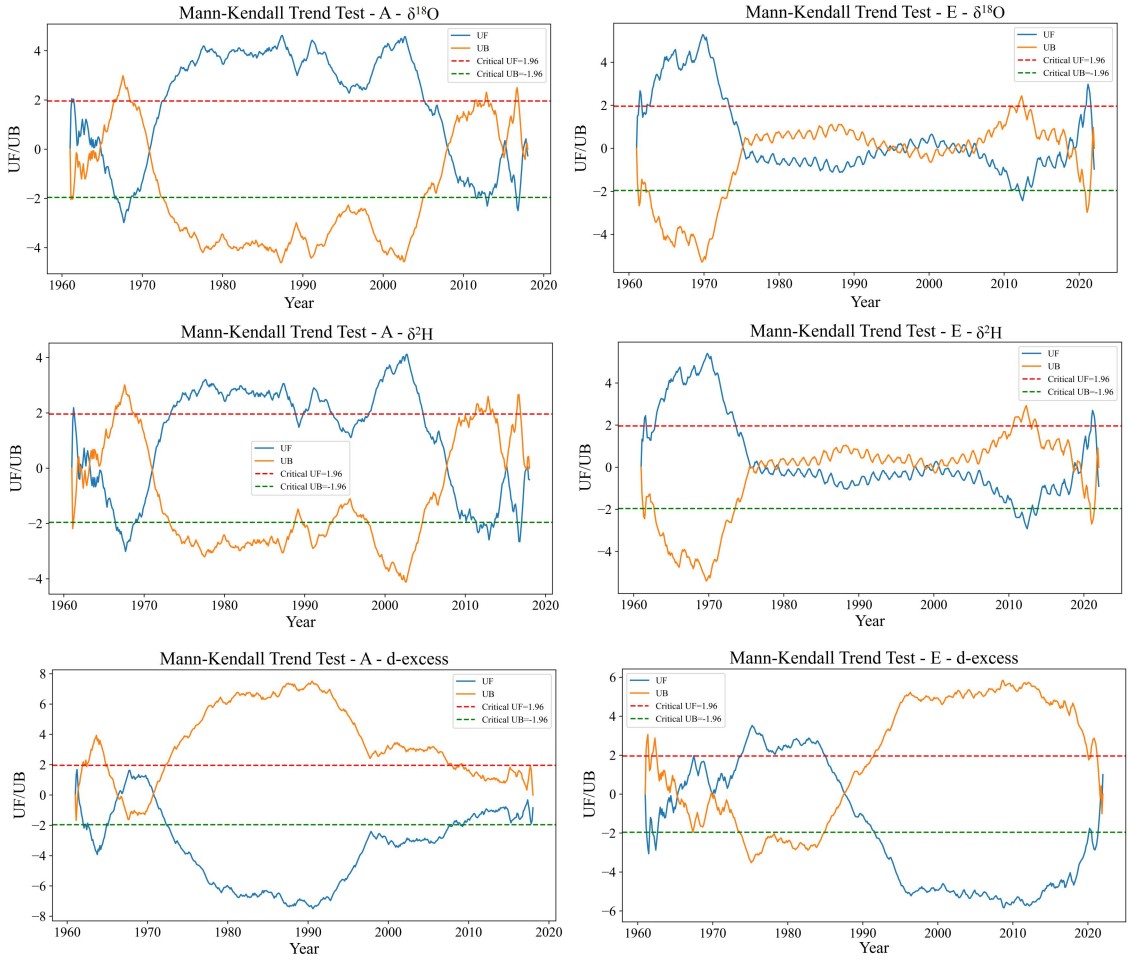

**Fig.3** Time series MK test for temperate (C) and cold (D) climates

## 4. Results and discussion

4.1 Temporal and Spatial Variation Characteristics of Precipitation Stable Isotopes

On a temporal scale, stable isotopes in precipitation exhibit pronounced seasonal variations, with higher values during the summer and lower values during the winter (Figure 4). This is attributed to seasonal variations in evaporation caused by temperature changes, resulting in the evaporative fractionation of stable isotopes in precipitation. Considering the completeness of the time series and regional differences within the Eurasian continent, we constructed a time series of precipitation stable isotopes based on the Köppen climate classification "climate zones" The temporal changes in precipitation stable isotopes under different climate types show significant differences. In tropical climates (A), the values of precipitation stable isotopes are higher, with low values reflecting enhanced precipitation. The "precipitation effect" in the Eurasian continent is particularly significant in tropical climates

(Tharammal et al., 2017), and the composition of precipitation stable isotopes reflects the correlated changes between temperature and precipitation. However, the seasonal fluctuations in tropical precipitation stable isotopes are minimal, and there is a fluctuating trend over approximately 20 years. Most arid climates (B) and temperate climates (C) on the Eurasian continent are under the influence of the westerly system. Before 1980, temperate climates experienced significant fluctuations in precipitation stable isotopes, followed by a stable period of about 30 years. After 2010, an unstable trend has become more pronounced, reflecting an increase in extreme weather events (Yao et al., 2021; Zhang et al., 2012). In arid climate regions, precipitation stable isotopes have undergone significant decreases. The Central Asian arid region is a typical temperate arid region, and numerous studies have pointed out a "warm and humid" trend in the climate of this region (Wang et al., 2020; Yan et al., 2019). The strengthening of the West Pacific subtropical high, North American subtropical high, and the Asian subtropical westerly jet is believed to increase precipitation in this region (Chen et al., 2011). The enhancement of high-latitude water vapour transport is a major factor influencing the increase in precipitation in the Central Asian arid region, which is also the reason for the decreasing trend in deuterium excess (Fig. 4, c-1). Cold climates (D) and polar climates (E) have the smallest values of precipitation stable isotopes, but they exhibit significant differences on an annual scale and a gradually increasing trend on an interannual scale. With global warming, high-latitude regions will provide more sources of water vapour for the water cycle (Ding et al., 2017).

On a spatial scale, the topographic differences and latitude variations in the region are the primary causes of spatial differences in stable isotopes in precipitation across the Eurasian continent. The multi-year average values of $\delta^2H$ and $\delta^{18}O$ at different latitudes are as follows: from 0° to 30°N, they are -30.20‰ and -5.99‰, from 30° to 60°N, they are -58.94‰ and -8.77‰, and from 60° to 90°N, they are -92.98‰ and -12.69‰. The Alps and the Tibetan Plateau form regions of low precipitation stable isotopes that differ from those at the same latitudes. The gradual uplift of the Tibetan Plateau's mountains leads to changes in the atmospheric circulation patterns over a larger area, altering the inherent characteristics of water vapour source regions, vapour transport paths, and precipitation stable isotope values. The response of precipitation stable isotopes to the plateau's climate reflects changes in the

large-scale circulation state (Yao et al., 2013). The isotopic variations in the surrounding
regions of the Alps reflect differences in water vapour sources due to regional topography
(Natali et al., 2021; Rindsberger et al., 1983). Spatial variations in deuterium excess can
effectively reflect differences in regional water vapour sources, with average values of
approximately 10‰ for tropical and temperate climates. Cold climate regions have lower
deuterium excess values, and due to the overlap of arid climates with other climate zones, the
distribution range of deuterium excess values in arid climates is larger. Therefore, it can be
hypothesized that if isotope-related variables (e.g., d-excess) are included in climate zone
classification criteria, more climate zones influenced by circulation patterns could be
identified.

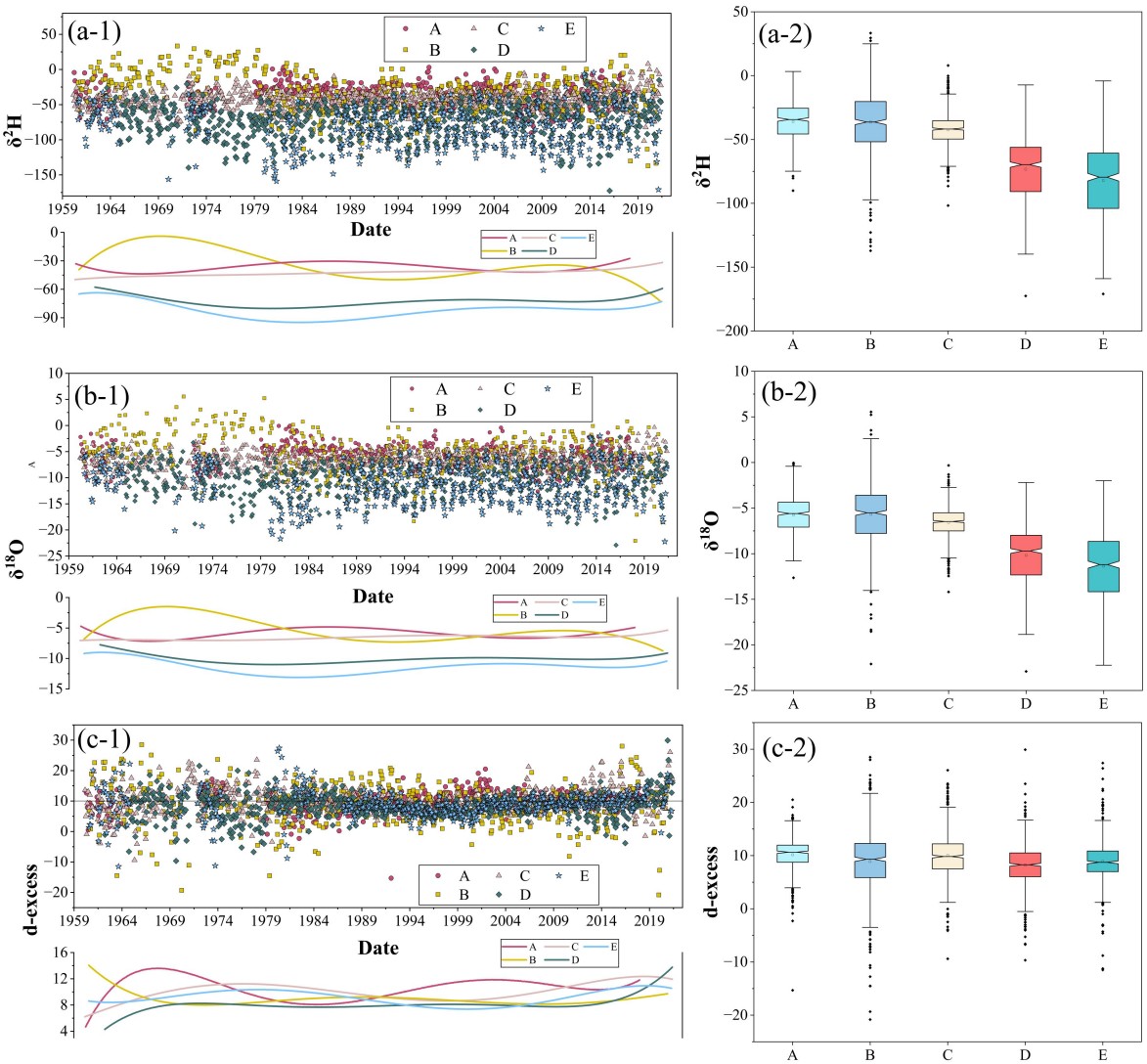


**Fig.4** The time series variations of $\delta^2$H, $\delta^{18}$O, and d-excess in the Eurasian continent.

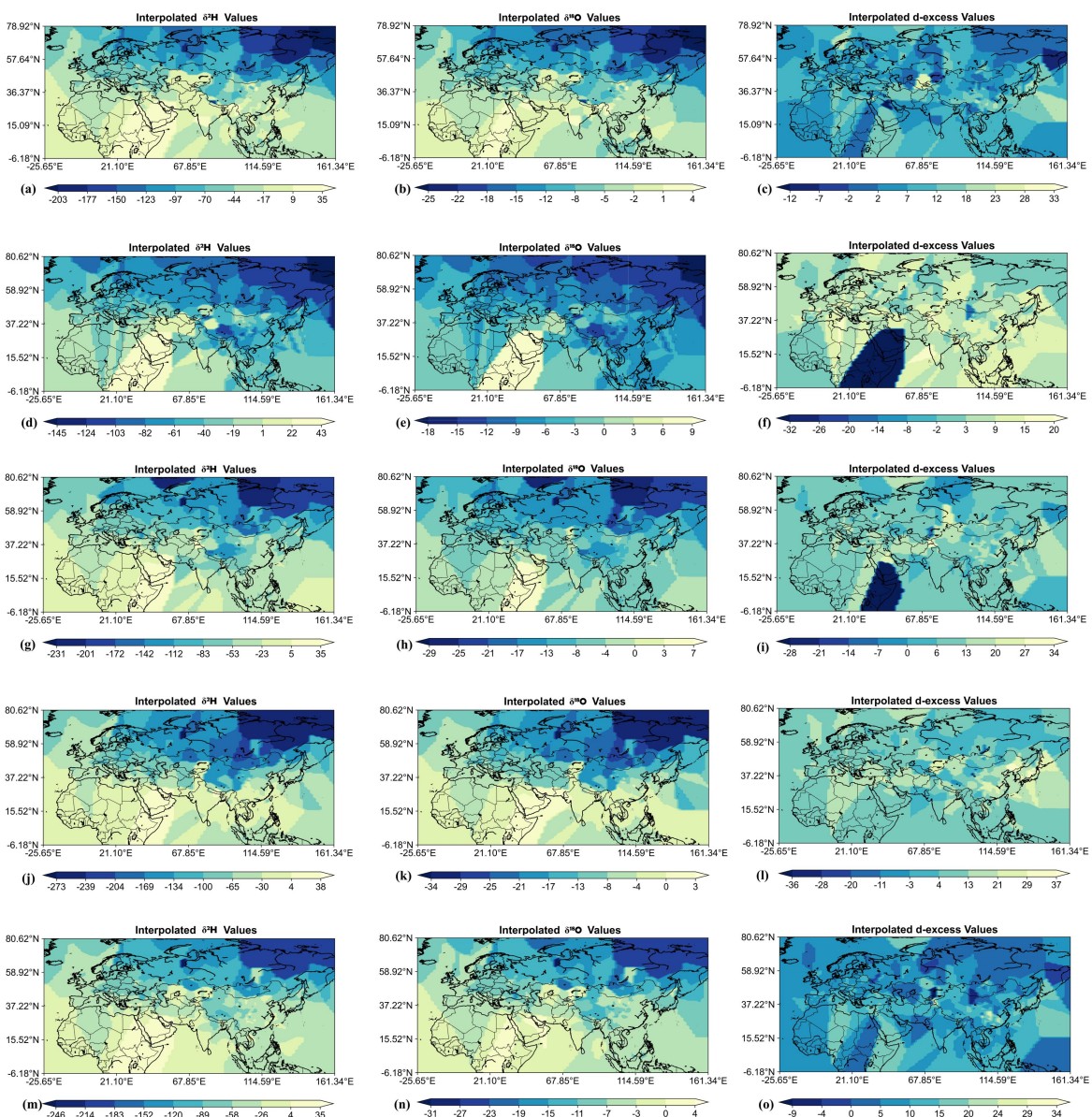

**Fig.5** The spatial variations of δ²H, δ¹⁸O, and d-excess in the Eurasian continent. Panels (a), (b), and (c) display the spatial distribution of isotope values in the spring season. Panels (d), (e), and (f) show the spatial distribution of isotope values in the summer season. Panels (g), (h), and (i) present the spatial distribution of isotope values in the autumn season. Panels (j), (k), and (l) exhibit the spatial distribution of isotope values in the winter season. Panels (m), (n), and (o) display the spatial distribution of isotope values averaged over multiple years.


4.2 Seasonal changes in meteoric water line and precipitation stable isotopes

The temporal and spatial variations of stable isotopes in precipitation are greatly

influenced by meteorological factors, and the changes in precipitation isotopes are consistent
with the climatic regions. Therefore, based on the Köppen climate classification, we
performed climate zoning for stable isotopes in precipitation sites. We used the least squares
method to fit meteoric water line for different climate regions (Fig.6) and considered the
seasonal variations of precipitation stable isotopes in different climate regions (Fig.7). The
meteoric water line for different climate types indicate relatively small differences in various
climate precipitation amounts in tropical climates. The variations in the slope and intercept of
the meteoric water line are determined by the combined effects of precipitation and
temperature, with convective precipitation weakening the impact of the "temperature effect."
Intense convective rainfall and oceanic water vapour transport bring abundant precipitation to
tropical regions. The fractionation mechanisms and variations of precipitation stable isotopes
not only reveal the inherent patterns of weather pattern occurrence and development (Sun et
al., 2022) but also correlate weather patterns with supply sources, tracing the water sources of
surface water bodies (Scholl and Murphy, 2014; Anon, 2017). Stable isotopes in precipitation
in arid climates are influenced by secondary evaporation below clouds, and intense
unbalanced fractionation processes lead to relative enrichment of stable isotopes in
precipitation (Wang et al., 2021; Zhu et al., 2021). Water resources are the most limiting
factor for the ecological and social environment in arid climate regions (García-Ruiz et al.,
2011). Therefore, compared to other climate regions, water recovery becomes more critical.
Stable isotopes in precipitation can accurately quantify water recovery and effectively assess
the impact of evaporation on different water bodies in arid regions. The majority of the global
population is distributed in temperate regions. Therefore, with global temperature rise, the
climate change situation in temperate regions deserves more attention. In temperate climate
zones, the differences in stable isotope composition between different climate types become
more significant. In the Mediterranean region controlled by the Summer Dry Warm Climate,
the slope and intercept are the lowest, indicating that temperature rise dominates the
fractionation of hydrogen and oxygen stable isotopes in precipitation, and the region shows a
trend of aridification under long-term average conditions. The westerly system is the main
controlling circulation in this region, and the changes in precipitation stable isotopes reflect
the attenuation trend of mid-latitude westerly moisture inward migration (Zhu et al., 2023; Shi
et al., 2021). In polar climates, the atmospheric water line exhibits higher slope and intercept.
The influence of unbalanced fractionation processes after water vapour condensation in cloud
systems is relatively small, resulting in a slope close to 8.
The seasonal variation of hydrogen and oxygen stable isotopes in precipitation on the
Eurasian continent generally exhibits a pattern of higher values in summer and lower values
in winter (Fig.7)(Hydrogen isotopes ($\delta^2$H) in Supplement S3). However, there are still
significant differences under different climate zones. The seasonal differences in tropical
climates are less pronounced, with the Tropical Sparse Forest Climate (Aw) showing a
decrease and increase with the months, possibly due to an increase in precipitation. Temperate
and cold climates generally exhibit significant seasonal variations. The deuterium excess in
the Eurasian continent shows a lower pattern in summer and a higher pattern in winter,
indicating seasonal changes in water vapour sources and transport distances (Zhang et al.,
2021a). This overall suggests that the summer climate in Eurasia is more humid, while the
winter climate is drier. Deuterium excess usually indicates the degree of imbalance in
seawater sources during their evaporation process, and it typically depends only on the
environmental conditions of the evaporation source. Compared to $\delta^2$H and $\delta^{18}$O, deuterium
excess displays a more stable pattern and is distributed around the global average (10‰). The
westerly and monsoon systems are the primary atmospheric circulation systems over the
Eurasian continent, carrying water vapour from the ocean inland and gradually weakening.
This indicates that the humidity in the vast region of Eurasia is strongly influenced by ocean
water vapour. Ocean conditions and large-scale atmospheric circulation changes can have
profound effects on the climate environment of the Eurasian continent.
The differences in precipitation stable isotopes among different climate types are not
only responses to different climate characteristics but also provide effective tools for a deeper
understanding of the process, climate change mechanisms, water vapour transport between
land and sea, and supply relationships between water bodies. The precipitation stable isotopes
dataset we have constructed for the Eurasian continent can be combined with traditional
meteorological data to provide more information for climate and hydrological research.

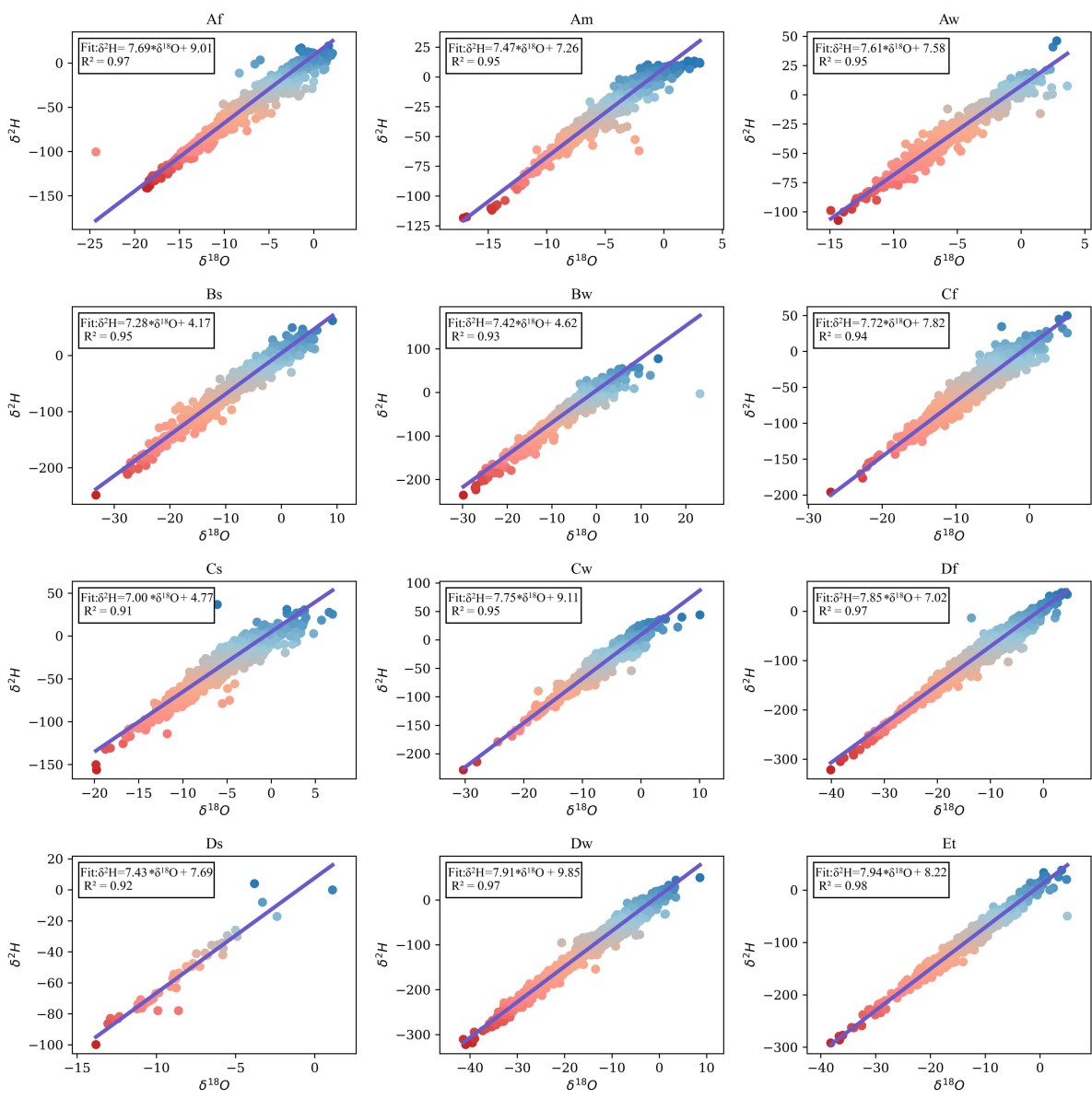

**Fig.6** Different meteoric water lines in various climate zones.


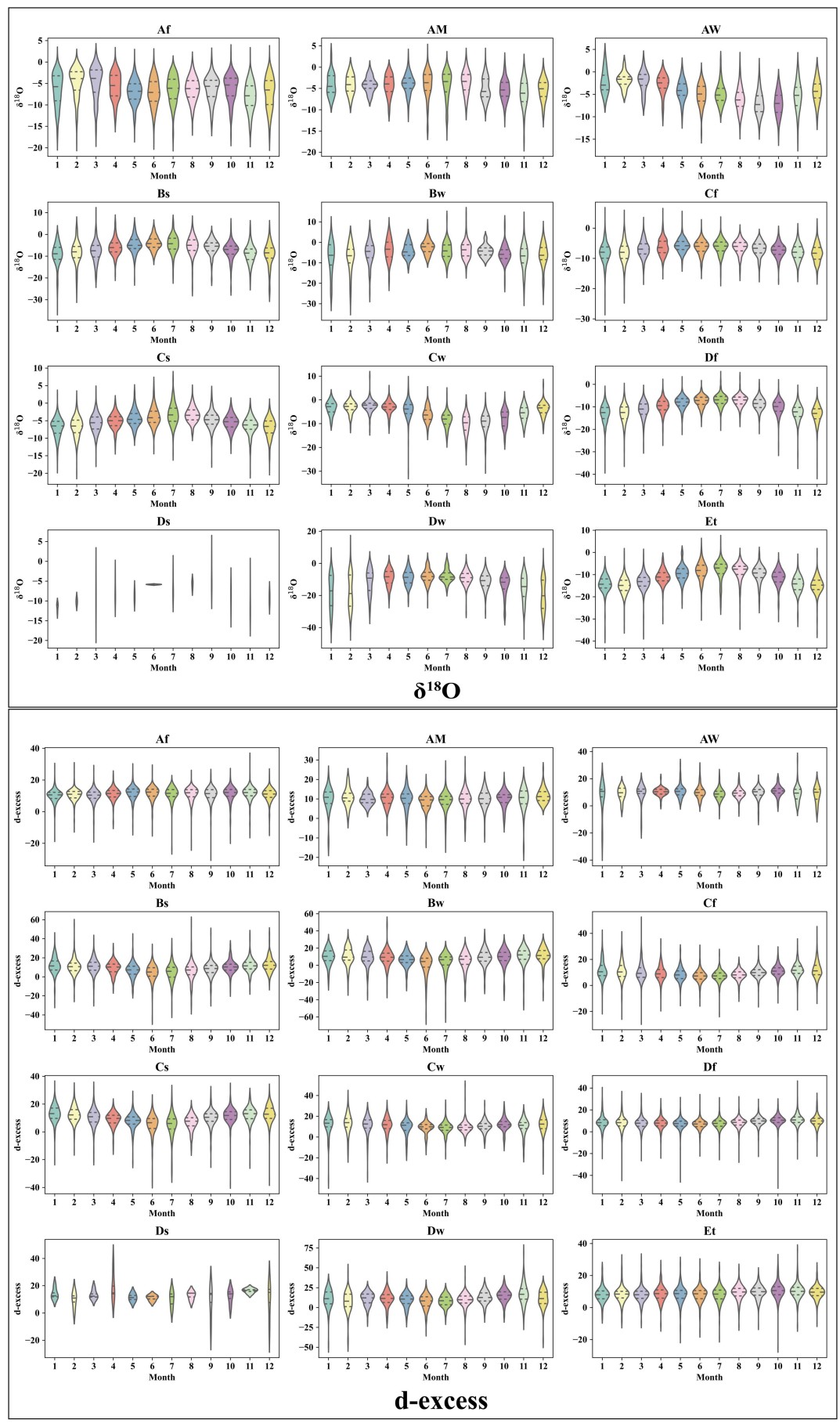

**Fig.7** Seasonal Distribution and Variations of Stable Isotopes in Precipitation ($\delta^{18}$O, d-excess)
4.3 Drivers of stable isotope variation in precipitation in Eurasia
Meteorological variables accompany the fractionation process of stable hydrogen and
oxygen isotopes in precipitation, impacting the composition of stable isotopes (Sun et al.,
2019). We utilized a random forest regression model to assess the importance of
meteorological variables in the Eurasian continent on stable isotopes. Random forest
regression is a non-parametric method used to solve prediction problems. It predicts
regression problems based on the average results of random decision trees, which use
bootstrapping to eliminate the possibility of overfitting (Erdélyi et al., 2023). The random
forest regression analysis of the fitted stable isotopes of hydrogen and oxygen showed good
goodness of fit for both the training and testing sets, indicating that temperature, precipitation,
potential evapotranspiration, vapour pressure, wind speed, and relative humidity have a high
explanatory power for stable isotopes of hydrogen and oxygen (Fig.8). The results of
cross-validation for the model indicate superior predictive performance for the target variable
$\delta^{18}$O compared to the target variable $\delta^2$H, as reflected in the smaller root mean square error
(RMSE) and mean absolute error (MAE) for $\delta^{18}$O (Supplement, Table S3). The composition
of stable isotopes in precipitation is greatly influenced by meteorological variables. Among
the six variables considered, temperature has the strongest explanatory power for the variation
of stable isotopes of hydrogen and oxygen, and potential evapotranspiration also has a
relatively strong explanatory ability, indicating that temperature change primarily drives the
variation of stable isotopes in precipitation in the Eurasian continent. The relative humidity is
the ratio of actual vapour pressure to saturated vapour pressure, but there is a significant
difference in the explanatory power of vapour pressure and relative humidity on stable
isotopes. Vapour pressure has a wider range of variation in the atmosphere, thus it may have
greater variability in the regression model, leading to a larger impact on predicting stable
isotopes in precipitation. Relative humidity, on the other hand, is a relative indicator with a
relatively smaller range of variation, so it may have a weaker predictive ability for stable
isotopes in precipitation in the regression model. The driving factors for the variation of stable
isotopes in precipitation in the Eurasian continent include climate change, seasonal variations,
topography and landforms, as well as water cycle processes, which collectively influence the

isotopic composition of precipitation. Atmospheric circulation directly affects the source of water vapor and the path of moisture, while other factors primarily influence the composition of stable isotopes in precipitation by altering temperature. For example, potential evapotranspiration plays a crucial role in explaining the variation of stable isotopes in precipitation. However, the control of meteorological variables on stable isotopes in precipitation varies between regions. Studies on two precipitation stations in Crimea have shown weak correlations between temperature, precipitation, and stable isotopes in precipitation. The complex natural environment determines that no single factor has a dominant control over the stable isotopes in precipitation in that region, and the composition of stable isotopes in precipitation is influenced by both local and distant factors (Dublyansky et al., 2018). In the eastern coastal region of China, the relative enrichment of stable isotopes in precipitation is due to the proximity to the evaporative source of the ocean, leading to an increased abundance of heavy isotopes (Zhang et al., 2021b). In the arid region of central Asia, there is a strong correlation between stable isotopes in precipitation and temperature, and the enrichment or depletion of stable isotopes in precipitation reflects the trend of temperature change (Zhu et al., 2023). In summary, the meteorological control factors of the composition of stable isotopes in precipitation vary in different regions. There is a strong relationship between stable isotopes in precipitation and meteorological variables, and stable hydrogen and oxygen isotopes may be considered essential climate response variables, which will contribute to describing the hydrological cycle and better predicting the response of future climate change and ecosystem changes.

Stable isotopes in precipitation, serving as indicators of climate and environment, play a unique role in enhancing the process-oriented understanding of extreme weather events and exploring hydrological connections between different water bodies. However, a limitation remains in the insufficient observation of stable isotopes in precipitation. Therefore, isotope atmospheric circulation models based on physical mechanisms have been widely applied to predict stable isotopes in water (Risi et al., 2012; Bowen et al., 2019). Physical models with different driving mechanisms can meet various usage needs, including paleoclimate reconstruction. For example, CAM3 simulation outputs precipitation oxygen isotope data (Lin et al., 2024). Machine learning is a novel approach for predicting stable isotopes in

precipitation, and European simulation practices indicate that oxygen isotope simulations
have shown good results, while simulations for hydrogen isotopes remain challenging (Nelson
et al., 2021). In general, uncertainties in both physical models and machine learning need
continuous improvement and refinement through real-world data. Additionally, an accurate
understanding of the influencing factors of stable isotopes in precipitation is fundamental for
achieving successful predictions through machine learning.

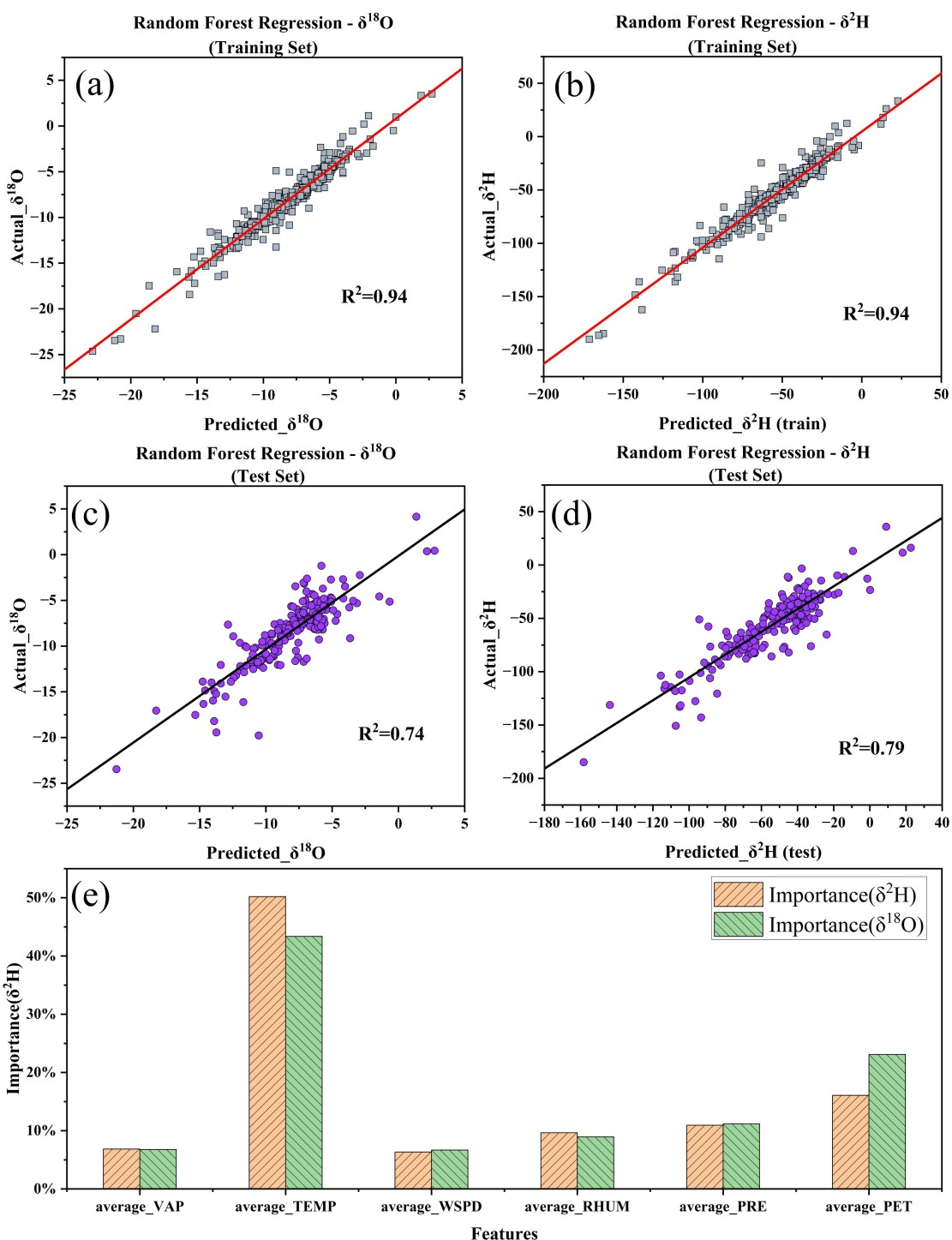

**Fig.8** Results of Random Forest Regression Analysis for $\delta^2H$ and $\delta^{18}O$ in Relation to Meteorological Variables.a: Regression results for the training set of $\delta^{18}O$, b: Regression results for the training set of $\delta^2H$, c: Regression results for the testing set of $\delta^{18}O$, d: Regression results for the testing set of $\delta^2H$, e: Importance of meteorological variables for $\delta^2H$ and $\delta^{18}O$.

## 5. Summary and outlook

Stable isotopes in precipitation play a crucial role in both the climate and hydrological

systems, exhibiting sensitivity to variations in both time and space. Research indicates significant differences in isotopic values between summer and winter, correlating with seasonal changes in temperature and evaporation. The temporal and spatial variations of precipitation stable isotopes vary significantly across different climate types, reflecting the influence of climate characteristics on isotopic distribution. Terrain and latitude differences are the primary reasons for spatial variations in stable isotopes in precipitation. Meteorological factors have a notable impact on precipitation stable isotopes, as evidenced by the meteoric water line in different climate types, revealing the influence of climate on isotopic fractionation. Observations of precipitation stable isotopes contribute to understanding weather patterns, water vapour sources, and transport pathways, providing important insights into stable isotope variations in arid climates. The integrated dataset of stable isotopes in precipitation from the Eurasian continent that we have compiled can offer more detailed climate and hydrological information. However, future research efforts should focus on improving observational data for Stable isotopes in precipitation. The uncertainties in physical models and machine learning methods need refinement through additional real-world data to enhance the accuracy of predicting precipitation stable isotopes.

**Data Availability**

Zhu, Guofeng (2024), "Dataset of Stable Isotopes of Precipitation in the Eurasian Continent", Mendeley Data, V2, doi: 10.17632/rbn35yrbd2.2

**Author Contribution Statement**

Longhu Chen: Conceptualization and Writing-Original draft preparation; Qinqin Wang: Writing and Data processing; Guofeng Zhu: Writing review and editing; Xinrui Lin: Modification; Dongdong Qiu: Modification; Yinying Jiao: data processing; Siyu Lu: Experiment; Rui Li: Methodology; Gaojia Meng: Visualization; Yuhao Wang: Visualization.

**Declaration of Interest Statement**

The authors declare that they have no known competing financial interests or personal relationships that could have appeared to influence the work reported in this paper.

**Acknowledgements**

This research was financially supported by the National Natural Science Foundatio

n of China(42371040, 41971036), Key Natural Science Foundation of Gansu Province
(23JRRA698), Key Research and Development Program of Gansu Province(22YF7NA1
22), Cultivation Program of Major key projects of Northwest Normal University(NWN
U-LKZD-202302), Oasis Scientific Research achievements Breakthrough Action Plan Pr
oject of Northwest Normal University(NWNU-LZKX-202303). The authors thank their
Northwest Normal University colleagues for their help in fieldwork, laboratory analysis,
and data processing.

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
