# Peer review of "Dataset of Stable Isotopes of Precipitation in the Eurasian Continent"

_Earth System Science Data, 2023_

## Referee Comment (RC1)

**General comments:**

The manuscript titled "Dataset of Stable Isotopes of Precipitation in the Eurasian Continent" (essd-2023-384) effectively compiles stable isotope data from various sources across the Eurasian continent. The data amassed by the authors' research group is notably unique and invaluable. They have meticulously analyzed the meteorological factors that influence stable isotopes in precipitation, considering the diverse climatic conditions across the Eurasian continent. After a thorough review of the data, I am convinced that the quality control measures for the stable isotope data are stringent, rendering it the most comprehensive dataset of its kind globally. The release of this dataset promises to significantly advance research in hydrology and meteorology, particularly concerning stable isotopes in precipitation. I also commend the manuscript for its clear structure and articulate writing style, and I wholeheartedly endorse its publication.

Nevertheless, there are certain aspects that require attention prior to publication. Specifically, the manuscript would benefit from a more detailed explanation of the mechanisms governing the spatial distribution of isotopes. Additionally, the presentation and argumentation within the manuscript could be further refined. Thus, I recommend that the article be considered for publication once these issues are addressed.

**Major comments:**

1. In the introductory section, the authors address the determinants of stable isotopes in precipitation and their relevance in various domains. However, some arguments lack comprehensiveness and should be meticulously revised and expanded to enhance clarity and thoroughness.

2. In the results and discussion section, the depiction of the connections between meteorological variables and stable isotopes in precipitation could be more precise. Numerous elements affect stable isotopes in precipitation, including water vapour transport, phase transitions of water, and evaporation beneath cloud cover. These elements are reflected in broad-scale atmospheric circulation and local geographical

disparities. It is crucial to avoid overly simplifying the impact of individual factors. For instance, in line 268, the discussion is limited to the effect of temperature on stable isotopes in precipitation, which is inadequate.

**Specific comments:**

1. In line 35, the reference to "State of the Climate in Asia 2022" is mentioned. It is important to provide specific citations for this reference in the subsequent text.

2. In lines 54-55, the phrase "spatial 'elevation' and 'latitude' effects" should be revised to "elevation effects".

3. In lines 85-88, it is indeed true that observed data are generally more precise than model validations. However, the role of models should not be overlooked, especially on a global scale. It is important to express this in a balanced and reasonable manner.

4. In line 144, for referencing the data, the citation link should be placed after "Water Isotope Network" for better clarity and organization.

5. In line 180, it is suggested to include the assessment of normality distribution for the data, along with the various statistical tests conducted. This will provide a more comprehensive analysis of the dataset.

6. In lines 189-190, it is mentioned that abnormal values were detected in the data. How were these abnormal values handled?

7. In section 4.1, there is a further discussion on the role of extreme stable isotopes of precipitation. It is suggested to consider adding additional content for discussion.

8. In line 209, it is stated that there is an increase in extreme events in the Eurasian continent. It is recommended to include supporting literature references for this argument.

9. In section 4.2, there are several abbreviations for climate types used in the figures to convey important information concisely and effectively. It is suggested to use the full names of the climate types in the text of the article to enhance the readers' understanding of the content.

10. In line 240, the sentence "The substantial reduction..." is incomplete and should be

clarified for better expression.

11. In line 254, when referring to the Köppen climate classification, it is recommended to cite the original references that established this classification system.

12. In line 286, how should we interpret the phrase "more stable pattern"?

13. In line 286, it is suggested to clearly state the range of the influence of atmospheric circulation and oceanic moisture on stable isotopes of precipitation in different regions of the Eurasian continent, as reflected in the data. This will help strengthen the argument and make it more comprehensive.

14. In section 4.3, it is stated that regional differences in stable isotopes of precipitation are attributed to regional differences in meteorological factors. The examples cited from the three references should be elaborated on in detail, specifically relating them to the content of this article.

15. In line 301, the term "Asia and Europe" is used to refer to the Eurasian continent in the title. It is recommended to use consistent professional terminology throughout the entire text.

16. In line 314, how should we interpret the phrase "the resampling process of"? It should be further explained.

17. It is recommended that the author provide more detailed definitions and descriptions for complex concepts and processes in order to help readers better understand them.

18. In the introduction, the article mentions the future application contributions of the dataset. It is suggested to explicitly state these potential scientific application contributions in the discussion section. This will help enhance the impact of your paper.

19. In Figure 6, "H" and "O" are used to represent variables instead of directly using "$\delta^2H$" and "$\delta^{18}O$". It is recommended to use the more specific notation of "$\delta^2H$" and "$\delta^{18}O$" to avoid confusion and ensure clarity in representing the variables.

20. In Figure 8, the representation of the isotope is given as "$^{18}O$" instead of "$\delta^{18}O$". It is advisable to clearly indicate the notation as "$\delta^{18}O$" to avoid any misinterpretation by the readers.

---

## Author Response (AR1)

**Author responses**

We appreciate the constructive comments and suggestions provided by the editor and reviewers for our manuscript titled "**Dataset of Stable Isotopes of Precipitation in the Eurasian Continent**" with the reference **essd-2023-384**. In response to the reviewers' comments, we have diligently revised our manuscript. The revised sections have been highlighted in red in the tracked changes version of the manuscript. The primary corrections and our responses to the reviewers' comments are outlined below.

**Responses to the reviewer's comments**

Response to Reviewer #1

**General comments:**

**The manuscript titled "Dataset of Stable Isotopes of Precipitation in the Eurasian Continent" (essd-2023-384) effectively compiles stable isotope data from various sources across the Eurasian continent. The data amassed by the authors' research group is notably unique and invaluable. They have meticulously analyzed the meteorological factors that influence stable isotopes in precipitation, considering the diverse climatic conditions across the Eurasian continent. After a thorough review of the data, I am convinced that the quality control measures for the stable isotope data are stringent, rendering it the most comprehensive dataset of its kind globally. The release of this dataset promises to significantly advance research in hydrology and meteorology, particularly concerning stable isotopes in precipitation. I also commend the manuscript for its clear structure and articulate writing style, and I wholeheartedly endorse its publication.**

**Nevertheless, there are certain aspects that require attention prior to publication. Specifically, the manuscript would benefit from a more detailed explanation of the mechanisms governing the spatial distribution of isotopes. Additionally, the presentation and argumentation within the manuscript could be further refined. Thus, I recommend that the article be considered for publication once these issues are addressed.**

Reply: We would like to thank Reviewer 1 for the positive opinion about our work, for important comments and for the time spent on the paper reading and reviewing. We answered all concerns/comments and provided the detailed explanations below.

**Major comments:**

**In the introductory section, the authors address the determinants of stable isotopes in precipitation and their relevance in various domains. However, some arguments lack comprehensiveness and should be meticulously revised and expanded to enhance clarity and thoroughness.**

Reply: We have reorganized the introduction and results sections of the manuscript. We have expanded upon these arguments to ensure that readers can have a clearer understanding of the interrelationship between stable isotopes in precipitation and meteorological variables, along with their applications in relevant fields. Furthermore, we have elaborated on the significance of integrating the dataset of stable isotopes in precipitation.

The changes to the manuscript are as follows:

1. Introduction

[revised manuscript text omitted]

**In the results and discussion section, the depiction of the connections between meteorological variables and stable isotopes in precipitation could be more precise. Numerous elements affect stable isotopes in precipitation, including water vapour transport, phase transitions of water, and evaporation beneath cloud cover. These elements are reflected in broad-scale atmospheric circulation and local geographical disparities. It is crucial to avoid overly simplifying the impact of individual factors. For instance, in line 268, the discussion is limited to the effect of temperature on stable isotopes in precipitation, which is inadequate.**

Reply: We fully agree with your remarks regarding a more precise description of the relationship between meteorological variables and stable isotopes in precipitation. In particular, we acknowledge the deficiency pointed out in your comment on line 268, where only the influence of temperature on stable isotopes is discussed. We will address this limitation with significant additions in the discussion section to

comprehensively explore the impact of factors such as water vapor transport, phase transitions of water, and sub-cloud evaporation on stable isotopes. This approach ensures that our discussion delves more deeply without oversimplifying the influence of individual factors. We believe that with these additions, our paper will more comprehensively depict the relationship between meteorological variables and stable isotopes, providing a better reflection of the broad influences of atmospheric circulation and geographical differences. Simultaneously, it will reinforce the importance of integrating stable isotopes in precipitation data.

The changes to the manuscript are as follows:

[revised manuscript text omitted]

**Specific comments:**

**In line 35, the reference to "State of the Climate in Asia 2022" is mentioned. It is important to provide specific citations for this reference in the subsequent text.**

Reply: We indeed acknowledge the need to explicitly cite the relevant literature for the mention of "State of the Climate in Asia 2022" on line 35. In the revised manuscript, we will include specific citations to ensure that the references in the text align with the mentioned report.

The changes to the manuscript are as follows:

State of the Climate in Asia 2022: https://library.wmo.int/records/item/66314-state-of-the-climate-in-asia-2022, last access: 21 January 2024.

**In lines 54-55, the phrase "spatial 'elevation' and 'latitude' effects" should be revised to "elevation effects".**

Reply: The term "altitude effect" refers to the phenomenon where stable isotopes in precipitation gradually decrease with increasing altitude. Accurate expression of professional terminology is crucial for conveying the content of the paper precisely and facilitating reader understanding. However, based on the recommendations of the two reviewers, significant adjustments have been made to this paragraph. We have shifted the focus towards a combined discussion of the applications of stable isotopes in precipitation and the influencing factors. Emphasis has been placed on elucidating the significance of stable isotopes in precipitation in the fields of climate and hydrology. Consequently, we have removed the mentioned content. Additionally, we

have reviewed the entire manuscript for the accurate use of professional terminology to ensure the precision of the paper.

**In lines 85-88, it is indeed true that observed data are generally more precise than model validations. However, the role of models should not be overlooked, especially on a global scale. It is important to express this in a balanced and reasonable manner.**

Reply: We will emphasize the significance of observational data, highlighting its importance in accurately validating models at both global and regional scales. We will also underscore the role of models on a global scale. In discussing the interplay between models and observational data, we have elaborated on their mutual relationship.

The changes to the manuscript are as follows:

1. Introduction

On the other hand, using general atmospheric circulation models to simulate isotope circulation is a major method for comparing isotope distributions in precipitation under both modern and ancient conditions (Joussaume et al., 1984; Brady et al., 2019). Simultaneously, the comparison between simulated and observed precipitation stable isotopes provides valuable validation for the physical components of atmospheric circulation models (Joussaume et al., 1984; Ruan et al., 2019).

**In line 144, for referencing the data, the citation link should be placed after "Water Isotope Network" for better clarity and organization.**

Reply: We have adjusted the positioning of the links for citing data sources to ensure the accuracy of data references.

The changes to the manuscript are as follows:

The data collected are primarily from the Water Isotopes website (https://wateriso.utah.edu/waterisotopes/index.html) and the Global Network of Stable Isotopes in Precipitation (GNIP) operated by the International Atomic Energy Agency (IAEA).

**In line 180, it is suggested to include the assessment of normality distribution for the data, along with the various statistical tests conducted. This will provide a**

**more comprehensive analysis of the dataset.**

Reply: In the process of revision, we consider it unnecessary to conduct a normal distribution test. The primary purposes of a normal distribution test are as follows: (1) Inference of the applicability of statistical methods – many statistical methods (such as t-tests, analysis of variance, etc.) perform better when data approximates a normal distribution. Through normality tests, one can assess the suitability of applying these methods. (2) Accuracy of parameter estimation – many parameter estimation methods (such as maximum likelihood estimation) are more accurate when data approximates a normal distribution. If the data does not follow a normal distribution, alternative estimation methods may need to be considered. (3) Accuracy of hypothesis testing – in many statistical hypothesis tests, such as the residual normality test in regression analysis, it is required that the residuals of the data follow a normal distribution. If the residuals do not conform to a normal distribution, appropriate adjustments or non-parametric methods may need to be considered. Since our Mann-Kendall (MK) test (Fig.3) and box plots (Fig.4) do not depend on normal distribution, we have therefore removed this part of the content.

We believe the issue with this section lies in the perceived lack of rationale for the MK test. To address this, we have made the following modifications: We constructed time series of stable isotopes in precipitation for different regions based on climate types, analyzing trends and the distribution of stable isotopes in precipitation across different climate zones (Fig.4). Additionally, we performed the Mann-Kendall (MK) test on temperate (C) and frigid (D) climates (Fig.3). The reason for this choice is the relatively unclear boundaries between temperate, frigid, and arid zones.

The changes to the manuscript are as follows:

In addition, we selected the two climatic zones with the most significant differences, namely the tropical and polar zones. The reason for this choice is that the boundaries between temperate, frigid, and arid zones are relatively unclear, with subtle changes in trends. Mann-Kendall (MK) tests were conducted on the temporal variations of stable isotopes in precipitation for both climatic zones (Fig.3). For the tropical climate (A), the stable isotopes of precipitation ($\delta^2H$ and $\delta^{18}O$) exhibit multiple non-significant

periods of abrupt changes. There is a significant increasing trend from 1971 to 2005, followed by a non-significant decreasing trend since 2009. Overall, the deuterium excess (d-excess) shows a non-significant decreasing trend, but this trend has weakened after 1990. In the polar climate (E), there is a significant increasing trend before 1973, followed by non-significant periods of both increase and decrease after 1975. However, after 2010, a gradually significant increasing trend is observed. Since 1985, the deuterium excess has undergone a non-significant decreasing process, and after 2010, it gradually reaches a significant increasing trend. The uncertainty of the tests is mainly attributed to the spatiotemporal distribution and volume of the data.

[Figure]

**Fig.3** Time series MK test for temperate (C) and cold (D) climates

**In lines 189-190, it is mentioned that abnormal values were detected in the data. How were these abnormal values handled?**

Reply: Regarding the handling of outliers in the data collection, we implemented the following procedures:

(1) Data Cleaning: We conducted a cleaning process to identify and remove potential

outliers that may arise from recording errors or instrument malfunctions. This ensures that the foundational dataset for analysis is accurate and reliable.

(2) Statistical Methods: We employed statistical methods to identify data points that may deviate from the normal range. For data points identified as outliers, we conducted further validation and made appropriate adjustments in the analysis (Fig.4).

However, we did not address outliers in d-excess. This decision was made because d-excess is calculated based on stable isotopes of hydrogen and oxygen, potentially encompassing pertinent information.

[Figure]

**Fig.4** The time series variations of $\delta^2H$, $\delta^{18}O$, and d-excess in the Eurasian continent. **In section 4.1, there is a further discussion on the role of extreme stable isotopes of precipitation. It is suggested to consider adding additional content for discussion.**

Reply: We appreciate your suggestion, and in response, we have made substantial

revisions to strengthen the discussion in this section. Building upon the temporal and spatial variations of stable isotopes in precipitation across different climatic zones, we have elucidated the interplay of stable isotopes in precipitation in the context of meteorology and hydrology. Additionally, we have highlighted relevant applications of stable isotopes in precipitation.

The changes to the manuscript are as follows:

4.1 Temporal and Spatial Variation Characteristics of Precipitation Stable Isotopes

On a temporal scale, stable isotopes in precipitation exhibit pronounced seasonal variations, with higher values during the summer and lower values during the winter (Figure 4). This is attributed to seasonal variations in evaporation caused by temperature changes, resulting in the evaporative fractionation of stable isotopes in precipitation. Considering the completeness of the time series and regional differences within the Eurasian continent, we constructed a time series of precipitation stable isotopes based on the Köppen climate classification "climate zones " The temporal changes in precipitation stable isotopes under different climate types show significant differences. In tropical climates (A), the values of precipitation stable isotopes are higher, with low values reflecting enhanced precipitation. The "precipitation effect" in the Eurasian continent is particularly significant in tropical climates (Tharammal et al., 2017), and the composition of precipitation stable isotopes reflects the correlated changes between temperature and precipitation. However, the seasonal fluctuations in tropical precipitation stable isotopes are minimal, and there is a fluctuating trend over approximately 20 years. Most arid climates (B) and temperate climates (C) on the Eurasian continent are under the influence of the westerly system. Before 1980, temperate climates experienced significant fluctuations in precipitation stable isotopes, followed by a stable period of about 30 years. After 2010, an unstable trend has become more pronounced, reflecting an increase in extreme weather events (Yao et al., 2021; Zhang et al., 2012). In arid climate regions, precipitation stable isotopes have undergone significant decreases. The Central Asian arid region is a typical temperate arid region, and numerous studies have pointed out a "warm and humid" trend in the climate of this region (Wang et al., 2020; Yan et al., 2019). The strengthening of the

West Pacific subtropical high, North American subtropical high, and the Asian subtropical westerly jet is believed to increase precipitation in this region (Chen et al., 2011). The enhancement of high-latitude water vapour transport is a major factor influencing the increase in precipitation in the Central Asian arid region, which is also the reason for the decreasing trend in deuterium excess (Fig. 4, c-1). Cold climates (D) and polar climates (E) have the smallest values of precipitation stable isotopes, but they exhibit significant differences on an annual scale and a gradually increasing trend on an interannual scale. With global warming, high-latitude regions will provide more sources of water vapour for the water cycle (Ding et al., 2017).

On a spatial scale, the topographic differences and latitude variations in the region are the primary causes of spatial differences in stable isotopes in precipitation across the Eurasian continent. The multi-year average values of $\delta^2 H$ and $\delta^{18}O$ at different latitudes are as follows: from 0° to 30°N, they are -30.20‰ and -5.99‰, from 30° to 60°N, they are -58.94‰ and -8.77‰, and from 60° to 90°N, they are -92.98‰ and -12.69‰. The Alps and the Tibetan Plateau form regions of low precipitation stable isotopes that differ from those at the same latitudes. The gradual uplift of the Tibetan Plateau's mountains leads to changes in the atmospheric circulation patterns over a larger area, altering the inherent characteristics of water vapour source regions, vapour transport paths, and precipitation stable isotope values. The response of precipitation stable isotopes to the plateau's climate reflects changes in the large-scale circulation state (Yao et al., 2013). The isotopic variations in the surrounding regions of the Alps reflect differences in water vapour sources due to regional topography (Natali et al., 2021; Rindsberger et al., 1983). Spatial variations in deuterium excess can effectively reflect differences in regional water vapour sources, with average values of approximately 10‰ for tropical and temperate climates. Cold climate regions have lower deuterium excess values, and due to the overlap of arid climates with other climate zones, the distribution range of deuterium excess values in arid climates is larger. Therefore, it can be hypothesized that if isotope-related variables (e.g., d-excess) are included in climate zone classification criteria, more climate zones influenced by circulation patterns could be identified.

**In line 209, it is stated that there is an increase in extreme events in the Eurasian continent. It is recommended to include supporting literature references for this argument.**

Reply: We agree with your suggestion, and to substantiate the claim of an increase in extreme events over the Eurasian continent, it is necessary to refer to relevant scientific literature.

The changes to the manuscript are as follows:

Before 1980, temperate climates experienced significant fluctuations in precipitation stable isotopes, followed by a stable period of about 30 years. After 2010, an unstable trend has become more pronounced, reflecting an increase in extreme weather events (Yao et al., 2021; Zhang et al., 2012).

**In section 4.2, there are several abbreviations for climate types used in the figures to convey important information concisely and effectively. It is suggested to use the full names of the climate types in the text of the article to enhance the readers' understanding of the content.**

Reply: We agree that the inappropriate expression could lead to difficulties in comprehension during the reading process. Therefore, we have opted to use specific climate terms for clarification.

The changes to the manuscript are as follows:

In temperate climate zones, the differences in stable isotope composition between different climate types become more significant. In the Mediterranean region controlled by the Summer Dry Warm Climate, the slope and intercept are the lowest, indicating that temperature rise dominates the fractionation of hydrogen and oxygen stable isotopes in precipitation, and the region shows a trend of aridification under long-term average conditions.

However, there are still significant differences under different climate zones. The seasonal differences in tropical climates are less pronounced, with the Tropical Sparse Forest Climate (Aw) showing a decrease and increase with the months, possibly due to an increase in precipitation.

**In line 240, the sentence "The substantial reduction..." is incomplete and should**

**be clarified for better expression.**

Reply: In the original manuscript, this explains that global warming will have a greater impact on high-latitude regions in the future. We also noted some unclear expressions in the article. In response, we have made significant modifications to certain sections to ensure logical and coherent presentation of the research content in the manuscript.

The changes to the manuscript are as follows:

Cold climates (D) and polar climates (E) have the smallest values of precipitation stable isotopes, but they exhibit significant differences on an annual scale and a gradually increasing trend on an interannual scale. With global warming, high-latitude regions will provide more sources of water vapour for the water cycle (Ding et al., 2017).

**In line 254, when referring to the Köppen climate classification, it is recommended to cite the original references that established this classification system.**

Reply: We based our study on the Köppen climate classification, specifically the improved Köppen climate classification map developed by Hylke E. Beck et al. According to their work, they state: "We present new global maps of the Köppen-Geiger climate classification at an unprecedented 1-km resolution for the present-day (1980–2016) and for projected future conditions (2071–2100) under climate change." Therefore, we believe that referencing their work in this manner is justified.

**In line 286, how should we interpret the phrase "more stable pattern"?**

Reply: The global average d-excess value for precipitation is 10‰. D-excess typically indicates the degree of imbalance during the evaporation process of seawater sources. It usually depends solely on the environmental conditions of the water source region. Throughout the entire process from water evaporation into the atmosphere to eventual condensation and becoming rainwater, this value remains constant. Therefore, when referring to a "more stable pattern," it means that the deuterium excess values are close to the global average.

The changes to the manuscript are as follows:

Compared to $\delta^2H$ and $\delta^{18}O$, deuterium excess displays a more stable pattern and is distributed around the global average (10‰). The westerly and monsoon systems are the primary atmospheric circulation systems over the Eurasian continent, carrying water vapour from the ocean inland and gradually weakening. This indicates that the humidity in the vast region of Eurasia is strongly influenced by ocean water vapour. Ocean conditions and large-scale atmospheric circulation changes can have profound effects on the climate environment of the Eurasian continent.

**In line 286, it is suggested to clearly state the range of the influence of atmospheric circulation and oceanic moisture on stable isotopes of precipitation in different regions of the Eurasian continent, as reflected in the data. This will help strengthen the argument and make it more comprehensive.**

Reply: On one hand, d-excess typically depends solely on the environmental conditions of the water source region. Throughout the entire process from water evaporation into the atmosphere to eventual condensation and becoming rainwater, this value remains constant. On the other hand, the westerly and monsoon systems are the major atmospheric circulation systems over the Eurasian continent. They transport water vapour from the ocean inland and gradually weaken, influencing vast regions of the Eurasian continent. Deuterium excess provides a relatively coarse range (Fig.5), but a more detailed range cannot be effectively determined.

**In section 4.3, it is stated that regional differences in stable isotopes of precipitation are attributed to regional differences in meteorological factors. The examples cited from the three references should be elaborated on in detail, specifically relating them to the content of this article.**

Reply: This section aims to emphasize the regional differences in the influencing factors of stable isotopes in precipitation. We believe that the cited cases sufficiently illustrate the diversity and complexity of the influencing factors of stable isotopes in precipitation. Therefore, we have not made any changes to this part of the content.

**In line 301, the term "Asia and Europe" is used to refer to the Eurasian continent in the title. It is recommended to use consistent professional**

**terminology throughout the entire text.**

Reply: Our corrections are as follows, and we will also check for similar issues throughout the entire manuscript.

4.3 Drivers of stable isotope variation in precipitation in Eurasia

**In line 314, how should we interpret the phrase "the resampling process of"? It should be further explained.**

Reply: For meteorological variables in the random forest regression model, we resampled reanalysis data based on the locations of isotope sites. However, due to the proximity of some site locations, it led to an increase in data duplication. Therefore, we removed duplicate values. Simultaneously, we have removed this part of the content from the manuscript.

**It is recommended that the author provide more detailed definitions and descriptions for complex concepts and processes in order to help readers better understand them.**

Reply: We will enhance the explanation of conceptual content and mechanisms in the revised manuscript to improve reader understanding and better highlight the advantages of our dataset. The modifications will primarily be reflected in the "Results and Discussion" section. For detailed changes, you can refer to the responses under "**Major comments:**" or consult the revised manuscript.

**In the introduction, the article mentions the future application contributions of the dataset. It is suggested to explicitly state these potential scientific application contributions in the discussion section. This will help enhance the impact of your paper.**

Reply: We will consider and incorporate the following content in the revision to explicitly highlight the potential scientific contributions of the dataset: Stable isotopes in precipitation data can provide information about the formation and transport processes of precipitation, aiding in the interpretation of precipitation sources, propagation paths, and their relationship with atmospheric circulation. This contributes to a better understanding of the dynamics and meteorological mechanisms of precipitation. Stable isotopes in precipitation offer a deeper insight into the water

cycle processes, such as the study of land-atmosphere recycled water vapour. Additionally, the dataset of stable isotopes in precipitation provides more inputs for climate models, enhancing the validation of model accuracy. Regarding the revised content for this section, we have provided responses under **"Major comments:"** for your reference.

**In Figure 6, "H" and "O" are used to represent variables instead of directly using "$\delta^2H$" and "$\delta^{18}O$". It is recommended to use the more specific notation of "$\delta^2H$" and "$\delta^{18}O$" to avoid confusion and ensure clarity in representing the variables.**

Reply: We have made modifications to the figures, using more specific symbols directly representing "$\delta^2H$" and "$\delta^{18}O$" to ensure a clear representation of variables and avoid confusion.

[Figure]

Fig.6 Different meteoric water lines in various climate zones.

**In Figure 8, the representation of the isotope is given as "¹⁸O" instead of "δ¹⁸O". It is advisable to clearly indicate the notation as "δ¹⁸O" to avoid any misinterpretation by the readers.**

Reply: We have re-generated the result images of the random forest model to ensure clarity and readability.

Modifications are as follows:

[Figure]

Response to Reviewer #2

**Stable isotope data in precipitation were compiled in this work starting from 1961 for various locations in Eurasia to facilitate potential applications (such as water resources and climate change studies). Analyses related to temporal and spatial variability and the key meteorological drivers for these isotope data were performed. This compilation of stable isotopes in precipitation can serve as – as discussed in the manuscript – an informative and useful dataset for the studies related to meteorology and climatology, hydrologic cycles, and long-term climate change. However, importantly, the discussions and analyses currently presented do not made it clear in terms of the additional benefits from these isotope datasets over the existing or more traditional/common meteorological and climatological data (especially given that these isotope data start from 1961), leading to the assessments on the scientific merit of this compilation/analysis work difficult. Please see my further comments for additional discussions. Therefore, some major revisions are recommended: including the discussions and analyses to highlight the significance of these isotope data, particularly emphasizing how these data can provide additional benefits over traditional/existing climate datasets.**

Reply: We would like to thank Reviewer 2 for the positive opinion about our work, for important comments and for the time spent on the paper reading and reviewing. We answered all concerns/comments and provided the detailed explanations below.

**Major comments:**

**Reading through the manuscript, it is unclear the how these stable isotope data provides additional benefits to end-users or scientific studies over the existing or more traditional datasets (such as reanalysis data as also mentioned and used in the paper). For example, these isotope data in this work start from 1961, which is not as early as some other available meteorological data such as reanalysis data (e.g., starting from 1940 in ERA5). Additionally, the isotope data is, in a way, a type of proxy data, which is affected by various parameters such as temperature and precipitation and therefore, use of isotope data to provide**

**information on temperature or precipitation is subject to a layer of complexity compared to the direct use of historical temperature and precipitation data. Consequently, one main question I have is how these isotope data provide additional benefits over the existing climate and meteorology data, perhaps the isotope data provides location-specific information? More emphases on the scientific and/or practical merit of these data are therefore needed.**

Reply: Traditional meteorological data, with its mature observation system and reanalysis methods, can effectively reveal meteorological and climatic information. However, stable isotope data in precipitation not only serves as a substitute indicator for traditional meteorological data but also has additional advantages compared to conventional meteorological datasets. In summary, stable isotope data in precipitation possesses the following advantages: enhanced process-oriented understanding of meteorological events (Aggarwal et al., 2016). Stable isotope data in precipitation can provide information about the formation and transport processes of precipitation, contributing to the interpretation of precipitation sources, propagation paths, and their relationships with atmospheric circulation. This aids in a better understanding of the dynamics and meteorological mechanisms of precipitation. Climate and environmental indicators: Precipitation isotope data can serve as climate and environmental indicators. They exhibit high sensitivity to climate change and ecosystem responses, making them valuable for studying the impacts of climate and environmental changes. Isotope data can also be used to trace pathways in the water cycle, providing insights into the sources and flow directions of water resources.

Stable isotopes in precipitation serve as a medium connecting the hydrological system and the climate system. Various water bodies, including seawater and land, serve as sources of water vapor, which, in turn, is the "substance source" of precipitation. Therefore, through the process of isotope fractionation in the water cycle, precipitation contains stable isotope markers from its initial sources. Stable isotope data in precipitation can be used to determine the sources and pathways of precipitation. By analyzing the isotopic composition of precipitation in different regions, it is possible to distinguish whether precipitation originates from oceanic or

land evaporation (precipitation from oceanic evaporation typically has higher $\delta18O$ and $\delta2H$). This contributes to understanding regional changes in atmospheric circulation, the migration and transport of water vapor. In addition, stable isotopes in precipitation provide a more in-depth understanding of the water cycle process. For example, in the study of terrestrial recycled water vapor, traditional meteorological data-based recycled water vapor models (box models) struggle to differentiate between advective water vapor and evaporative water vapor. In contrast, stable isotope-based recycled water vapor models can directly estimate changes in the contribution of recycled water vapor based on stable isotope sampling at precipitation sites (Zhang et al., 2021). Furthermore, they can differentiate contributions from evaporation and transpiration to precipitation. Stable isotope data in precipitation can enhance the accuracy of atmospheric circulation models. Integrating isotopes into atmospheric circulation models (e.g., GCM, LZMD, ICAM) provides valuable validation for the physical components of atmospheric circulation models through comparisons between simulated and observed stable isotopes in precipitation (Joussaume et al., 1984).

In the revised manuscript, adjustments have been made to the introduction, results, and discussion sections, and a "Summary and Outlook" section has been added. We have discussed the interconnections between stable isotopes in precipitation, climate, and hydrology, elucidating the practical applications of stable isotopes in precipitation. The importance of constructing a dataset of stable isotopes in precipitation has been emphasized.

The changes to the manuscript are as follows:

1. Introduction

[revised manuscript text omitted]

The temporal analyses carried out in this work (e.g., Figures 4 and 7) using the compiled isotope data seem to be based on the averages of all data locations in Eurasia, which to me are crude and potentially subject to limitations/bias. For example, in Figure 4, the temporal variability of the isotope data seems to be calculated as averages across all stations for each day (not clearly discussed in the text), which could be problematic depending on the periods of records in different stations: do all stations have a same period of record from 1961 to 2022? Is it possible that more stations/locations have more data during a recent period? If different historical periods have different amounts of stations/locations with available data, taking arithmetic means would inherently cause systematic bias. As this manuscript puts a particular emphasis on the spatial variability, these temporal analyses should at least be conducted for some smaller regions to be

**consistent. Such additional analyses may also be useful to demonstrate the merit of these station-level isotope data (related to my previous comment).**

Reply: We appreciate your concern regarding the use of integrated stable isotope data for temporal analysis. To address this issue, we have reanalyzed the data, computing time series on a regional basis (Köppen climate zones). The rationale for this division is as follows: the variation in stable isotopic composition of precipitation is significantly influenced by meteorological factors, and within the same climate zone, there is similarity in the variation of stable isotopic composition. To address the issue of site data having different historical periods, we have employed weighted averages. This approach assigns greater weight to sites with longer historical records, better reflecting the trends observed over the long term and mitigating the impact of short-term site data. This introduces a more equitable balance when dealing with data from sites with different historical periods, ensuring consistency with the spatial variability emphasized in our study.

The changes to the manuscript are as follows:

[Figure]

**Fig.4** The time series variations of δ²H, δ¹⁸O, and d-excess in the Eurasian continent.

**Minor comments:**

**Lines 14-15: in the main text of the manuscript, it will be helpful to summarize the dimensions of datasets (in addition to here in the abstract) such as a table showing the number of stations at different regions, the periods of records for each station, etc.**

Reply: We fully agree with the addition of a table to summarize the dimensions of the dataset, such as the number of sites in different regions, and the recording periods for each site.

Additionally, during the modification process, numerous redundant site counts for the same project were generated because we previously tallied quantities based on site names (some sites had different time samplings, leading to distinct "project IDs"). Now, we have recalculated the site quantity based on geographical coordinates (842

sites).

The modifications in the manuscript text are as follows:

**Table S1** Precipitation Stable Isotope Site Information

| Site_Name | Latitude | Longitude | Elevation (m) | Earliest Collection Date | Latest Collection Date | Data Source |
|---|---|---|---|---|---|---|
| 10-TG-16w | 32.9159 | 91.9596 | 5126.0000 | 2008.01.01 | 2008.01.01 | WaterIsotopes.org |
| 12QH18 | 34.4752 | 97.7328 | 4413.0000 | 2012.05.05 | 2012.05.05 | WaterIsotopes.org |
| A CORUNA | 43.3659 | -8.4214 | 58.0000 | 2000.02.15 | 2016.12.15 | GNIP |
| Aba | 31.9300 | 101.7200 | | 2021.07.06 | 2022.04.04 | Measured data |
| ADANA | 36.9800 | 35.3000 | 73.0000 | 1990.01.15 | 2016.12.15 | GNIP |
| ADZ SG RF | 1.2953 | 103.7711 | | 2015.03.06 | 2015.03.27 | WaterIsotopes.org |
| Akita | 39.4300 | 140.8000 | 10.0000 | 2013.01.27 | 2014.01.03 | WaterIsotopes.org |
| Akita-kita | 39.4800 | 140.2000 | 20.0000 | 2013.02.01 | 2014.01.07 | WaterIsotopes.org |
| Aktumsuk | 45.0918 | 59.3451 | | 2006.03.15 | 2006.03.15 | WaterIsotopes.org |
| ALEPPO | 36.1833 | 37.2167 | 410.0000 | 1989.12.15 | 1992.05.15 | GNIP |
| ALEXANDRIA | 31.1833 | 29.9500 | 7.0000 | 1961.10.15 | 2004.04.15 | GNIP |
| ALMERIA AEROPUERTO | 36.8464 | -2.3569 | 21.0000 | 2000.04.15 | 2016.12.15 | GNIP |
| ALOR STAR | 6.2000 | 100.4000 | 5.0000 | 1992.02.15 | 2016.12.15 | GNIP |
| ALTNABREAC | 58.3833 | -3.7000 | 155.0000 | 1981.08.15 | 1982.08.15 | GNIP |
| AMDERMA | 69.7667 | 61.6833 | 53.0000 | 1981.01.15 | 1989.09.15 | GNIP |
| AMMAN-WAJ | 31.9577 | 35.8483 | 900.0000 | 1999.01.15 | 2018.12.15 | GNIP |
| ANCONA-MONTE D'AGO | 43.5870 | 13.5153 | 170.0000 | 2000.10.15 | 2018.11.15 | GNIP |
| ANDALO | 46.1667 | 11.0000 | 1005.0000 | 1999.09.30 | 1999.09.30 | WaterIsotopes.org |
| ANIP, HZB Nr. 101287 | 47.3883 | 11.2631 | 990.0000 | 1993.01.15 | 2002.12.15 | WaterIsotopes.org |
| ANIP, HZB Nr. 102343 | 47.0000 | 11.5167 | 1450.0000 | 1991.01.15 | 2000.12.15 | WaterIsotopes.org |
| ANIP, HZB Nr. 103820 | 47.5961 | 13.1644 | 470.0000 | 2007.01.15 | 2013.12.15 | WaterIsotopes.org |
| ANIP, HZB Nr. 105734 (Schule) | 47.8040 | 13.7766 | 425.0000 | 1973.01.15 | 1983.12.15 | WaterIsotopes.org |

| | | | | | |
|---|---|---|---|---|---|
| ANIP, HZB Nr. 105908 | 47.3467 | 13.3950 | 910.0000 | 2007.01.15 | 2013.12.15 | WaterIsotopes.org |
| ANIP, HZB Nr. 110551 | 47.8517 | 16.8431 | 121.0000 | 2004.01.15 | 2013.12.15 | WaterIsotopes.org |
| ANIP, HZB Nr. 110569 | 47.7431 | 16.8433 | 119.0000 | 1990.01.15 | 1999.10.15 | WaterIsotopes.org |
| ANIP, HZB Nr. 113324, St. Peter i. Katschtal | 47.0283 | 13.5981 | 1220.0000 | 2007.01.15 | 2013.12.15 | WaterIsotopes.org |
| ANIP, HZB Nr. 113498 | 46.4896 | 14.5900 | 550.0000 | 2007.01.15 | 2013.12.15 | WaterIsotopes.org |
| ANIP, HZB Nr. 113886 Seeberg | 46.4225 | 14.5422 | 940.0000 | 2007.01.15 | 2013.12.15 | WaterIsotopes.org |
| ANIP, HZB Nr. 114173 Flughafen | 46.6484 | 14.3185 | 447.0000 | 2004.01.15 | 2013.12.15 | WaterIsotopes.org |
| ANIP, HZB-Nr. 100552 | 47.3115 | 10.0177 | 835.0000 | 2004.01.15 | 2013.12.15 | WaterIsotopes.org |
| ANIP, HZB-Nr. 100776 Rieden | 47.4886 | 9.7366 | 410.0000 | 2004.01.15 | 2012.12.15 | WaterIsotopes.org |
| ANIP, HZB-Nr. 101238 Reutte | 47.4833 | 10.7500 | 870.0000 | 2007.01.15 | 2013.12.15 | WaterIsotopes.org |
| ANIP, HZB-Nr. 101345 | 47.5333 | 11.7000 | 904.0000 | 1996.01.15 | 2002.12.15 | WaterIsotopes.org |
| ANIP, HZB-Nr. 102210 Obergurgl | 46.8667 | 11.0333 | 1940.0000 | 2007.01.15 | 2011.12.15 | WaterIsotopes.org |
| ANIP, HZB-Nr. 102236 | 47.0764 | 10.9700 | 1180.0000 | 2007.01.15 | 2013.12.15 | WaterIsotopes.org |
| ANIP, HZB-Nr. 102327 Flughafen | 47.2575 | 11.3539 | 580.0000 | 2007.01.15 | 2014.12.15 | WaterIsotopes.org |
| ANIP, HZB-Nr. 102418 Patscherkofel | 47.2088 | 11.4618 | 2245.0000 | 2005.01.15 | 2009.10.15 | WaterIsotopes.org |
| ANIP, HZB-Nr. 102814 | 47.5833 | 12.1667 | 495.0000 | 2010.01.15 | 2014.12.15 | WaterIsotopes.org |
| ANIP, HZB-Nr. 102939 | 47.2500 | 10.8833 | 695.0000 | 1994.10.15 | 2002.12.15 | WaterIsotopes.org |
| ANIP, HZB-Nr. 103853 Flughafen | 47.7942 | 13.0017 | 430.0000 | 2007.01.15 | 2013.12.15 | WaterIsotopes.org |
| ANIP, HZB-Nr. 104323 Braunau | 48.2539 | 13.0767 | 360.0000 | 2005.01.15 | 2013.12.15 | WaterIsotopes.org |
| ANIP, HZB-Nr. 105130 | 47.6000 | 13.7833 | 640.0000 | 1996.01.15 | 2002.12.15 | WaterIsotopes.org |
| ANIP, HZB-Nr. 105296 Feuerkogel | 47.8167 | 13.7239 | 1618.0000 | 2007.01.15 | 2013.12.15 | WaterIsotopes.org |

| | | | | | | |
|---|---|---|---|---|---|---|
| ANIP, HZB-Nr. 105684 | 47.9044 | 13.5722 | 469.0000 | 2007.01.15 | 2013.12.15 | WaterIsotopes.org |
| ANIP, HZB-Nr. 106542 Breitenau | 47.8464 | 14.3569 | 514.0000 | 2005.01.15 | 2013.12.15 | WaterIsotopes.org |
| ANIP, HZB-Nr. 106567 Planneralm | 47.4048 | 14.2009 | 1605.0000 | 2009.01.15 | 2013.12.15 | WaterIsotopes.org |
| ANIP, HZB-Nr. 107011 Lackenhof | 47.8700 | 15.1539 | 807.0000 | 2009.01.15 | 2012.09.15 | WaterIsotopes.org |
| ANIP, HZB-Nr. 107607 Ottenstein | 48.5836 | 15.3408 | 554.0000 | 2007.01.15 | 2013.12.15 | WaterIsotopes.org |
| ANIP, HZB-Nr. 107979 Hohe Warte | 48.2486 | 16.3563 | 198.0000 | 2005.01.15 | 2014.12.15 | WaterIsotopes.org |
| ANIP, HZB-Nr. 108456 | 47.8769 | 15.8934 | 495.0000 | 2007.01.15 | 2013.12.15 | WaterIsotopes.org |
| ANIP, HZB-Nr. 112599 Karlgraben | 47.6823 | 15.5627 | 775.0000 | 2008.01.15 | 2013.12.15 | WaterIsotopes.org |
| ANIP, HZB-Nr. 113001 | 46.7454 | 12.4078 | 1075.0000 | 2009.01.15 | 2013.12.15 | WaterIsotopes.org |
| ANIP, HZB-Nr. 113498 Villacher Alpe | 46.6036 | 13.6728 | 2120.0000 | 2007.01.15 | 2013.12.15 | WaterIsotopes.org |
| ANIP, HZB-Nr. 113522 | 46.6246 | 13.6867 | 907.0000 | 1993.01.15 | 2002.12.15 | WaterIsotopes.org |
| ANIP, HZB-Nr. 114280 St. Michael | 46.5586 | 14.7670 | 527.0000 | 1993.01.15 | 2002.12.15 | WaterIsotopes.org |
| ANIP, HZB-Nr. WEISSBACH | 47.5833 | 12.7000 | 629.0000 | 1983.01.15 | 1989.12.15 | WaterIsotopes.org |
| ANip, HZB-Nr.106252 | 47.6544 | 14.9809 | 610.0000 | 2007.01.15 | 2013.12.15 | WaterIsotopes.org |
| ANIP, Wasseralm HZB Nr. 110155 | 47.7350 | 15.6484 | 774.0000 | 1989.01.15 | 1998.12.15 | WaterIsotopes.org |
| ANKARA | 39.9500 | 32.8800 | 902.0000 | 1990.01.15 | 2016.12.15 | GNIP |
| ANTALYA | 36.8800 | 30.7000 | 49.0000 | 1990.01.15 | 2016.12.15 | GNIP |
| ANY | 35.7475 | 138.4166 | 526.0000 | 2010.07.15 | 2015.06.15 | WaterIsotopes.org |
| Aomori | 40.4900 | 140.4500 | 5.0000 | 2013.01.30 | 2014.01.01 | WaterIsotopes.org |
| AQABA | 29.5500 | 34.9000 | 2.0000 | 2002.01.15 | 2002.01.15 | GNIP |
| Arakawa | 35.4300 | 139.4500 | 20.0000 | 2013.02.06 | 2013.12.28 | WaterIsotopes.org |
| ARKHANGELSK | 64.5800 | 40.5000 | 13.0000 | 1981.02.15 | 1990.11.15 | GNIP |
| ARKONA | 54.6789 | 13.4342 | 42.0000 | 1997.06.15 | 2008.07.15 | GNIP |
| ARMAGH | 54.3533 | -6.6483 | 64.0000 | 2012.05.15 | 2019.12.15 | GNIP |

| OBSERVATORY | | | | | | |
|---|---|---|---|---|---|---|
| Artashat | 39.9322 | 44.5214 | 820.8000 | 2014.07.06 | 2015.04.28 | WaterIsotopes.org |
| ARTERN | 51.3744 | 11.2919 | 164.0000 | 1997.06.15 | 2013.12.15 | GNIP |
| ASTRAKHAN | 46.2500 | 48.0300 | -18.0000 | 1981.01.15 | 2000.12.15 | GNIP |
| AT_0007 | 47.7351 | 15.6486 | | 2005.06.07 | 2005.08.11 | WaterIsotopes.org |
| AT_0007_007 | 47.7351 | 15.6486 | | 2005.06.04 | 2005.06.04 | WaterIsotopes.org |
| AT_0141 | 47.7267 | 13.0354 | 510.0000 | 2012.05.25 | 2012.12.04 | WaterIsotopes.org |
| AT_0144 | 47.7215 | 12.9843 | 1396.0000 | 2012.05.25 | 2012.10.10 | WaterIsotopes.org |
| AT_1203 | 46.7156 | 13.0404 | 1391.0000 | 1995.01.01 | 1995.10.01 | WaterIsotopes.org |
| AT_1207 | 46.6762 | 12.9966 | 705.0000 | 1995.02.01 | 1995.09.01 | WaterIsotopes.org |
| AT_2088 | 48.5239 | 16.7556 | 205.0000 | 2009.01.15 | 2013.12.15 | WaterIsotopes.org |
| AT_2338 | 48.3297 | 14.2586 | 490.0000 | 2009.01.15 | 2013.12.15 | WaterIsotopes.org |
| AT_2402 | 47.0868 | 13.1151 | 1120.0000 | 1988.01.15 | 1997.12.15 | WaterIsotopes.org |
| AT_2433 | 47.0777 | 15.4489 | 366.0000 | 2004.01.15 | 2013.12.15 | WaterIsotopes.org |
| AT_2434 | 47.6409 | 13.8919 | 710.0000 | 2011.01.15 | 2013.12.15 | WaterIsotopes.org |
| AT_2704 | 48.1833 | 16.4000 | 156.4800 | 2015.01.23 | 2020.5.16 | WaterIsotopes.org |
| ATHENS-PENDELI | 38.0501 | 23.8667 | 498.0000 | 2001.01.15 | 2018.12.15 | GNIP |
| AVIGNON | 43.9128 | 4.8888 | 33.0000 | 1997.04.15 | 2018.12.15 | GNIP |
| Ayouqui weather station (AYQ) | 39.2200 | 101.6800 | 1620.0000 | 2013.07.12 | 2017.07.08 | WaterIsotopes.org |
| BAB-JANET | 35.5722 | 36.1897 | 1100.0000 | 1992.12.15 | 1993.04.15 | GNIP |
| BAD SALZUFLEN | 52.1042 | 8.7519 | 135.0000 | 1990.01.15 | 2011.12.15 | GNIP |
| BAGDARIN | 54.4667 | 113.5833 | 903.0000 | 1996.04.15 | 2000.10.15 | GNIP |
| BAHRAIN | 26.2700 | 50.6200 | 2.0000 | 1970.01.15 | 2018.10.15 | GNIP |
| Baiyin | 36.5400 | 104.1740 | | 2020.07.25 | 2021.07.02 | Measured data |
| BAKURIANI | 41.7333 | 43.5167 | 1665.0000 | 2008.03.15 | 2018.12.15 | GNIP |
| BALAS-BALAS | 9.2900 | 123.1700 | 827.0000 | 1997.01.15 | 1998.12.15 | GNIP |
| BALTI | 47.7000 | 27.8833 | 231.0000 | 2007.09.15 | 2015.08.15 | GNIP |

| | | | | | |
|---|---|---|---|---|---|
| BANDARBAN | 22.2000 | 92.2000 | 24.0000 | 2015.03.15 | 2018.09.15 | GNIP |
| BANGKOK | 13.7300 | 100.5000 | 2.0000 | 1990.01.15 | 2015.11.15 | GNIP |
| Baoji | 34.3693 | 107.1449 | | 2020.10.14 | 2021.11.10 | Measured data |
| Baoshan | 25.0800 | 99.1000 | | 2021.05.06 | 2022.01.14 | Measured data |
| BAOTOU | 40.6700 | 109.8500 | 1067.0000 | 1986.03.15 | 1992.10.15 | GNIP |
| Baotou | 40.6500 | 109.8300 | | 2021.05.13 | 2021.08.03 | Measured data |
| BARABINSK | 55.3333 | 78.3667 | 120.0000 | 1996.06.15 | 2000.12.15 | GNIP |
| BARI | 41.1333 | 16.8500 | 24.0000 | 2001.12.31 | 2001.12.31 | WaterIsotopes.org |
| BARISAL | 22.7000 | 90.3600 | 10.0000 | 2013.05.15 | 2018.10.15 | GNIP |
| Barisal | 22.7210 | 90.3520 | | 2013.05.06 | 2015.12.19 | WaterIsotopes.org |
| BASOVIZZA | 45.6439 | 13.8639 | 397.0000 | 1999.01.31 | 1999.01.31 | WaterIsotopes.org |
| BATTIPAGLIA | 40.6167 | 14.9667 | 71.0000 | 2001.05.31 | 2001.05.31 | WaterIsotopes.org |
| Bayan Nur | 40.7574 | 107.4170 | | 2021.05.22 | 2021.09.15 | Measured data |
| BAYIR | 30.7619 | 36.6769 | 902.0000 | 1968.11.15 | 1968.11.15 | GNIP |
| BEEK | 50.9200 | 5.7800 | 111.0000 | 1981.01.15 | 1992.04.15 | GNIP |
| BEER SHEVA | 31.2300 | 34.7800 | 270.0000 | 1965.11.15 | 1968.01.15 | GNIP |
| BELGAUM | 15.8806 | 74.4933 | 747.0000 | 2003.08.15 | 2005.10.15 | GNIP |
| BELLINZAGO | 45.5667 | 8.6333 | 190.0000 | 2000.12.31 | 2000.12.31 | WaterIsotopes.org |
| BERLIN | 52.4672 | 13.4019 | 48.0000 | 1990.01.15 | 2012.12.15 | GNIP |
| BERN | 46.9522 | 7.4393 | 511.0000 | 1990.01.15 | 2012.12.15 | GNIP |
| BET DAGAN | 31.9973 | 34.8162 | 39.0000 | 1984.01.15 | 2001.05.15 | GNIP |
| BEYROUTH | 33.8719 | 35.5097 | 19.0000 | 2003.11.15 | 2006.03.15 | GNIP |
| Biandianzhan | 37.3300 | 101.5100 | | 2021.11.27 | 2021.11.27 | Measured data |
| BLOUDAN | 33.7250 | 36.1303 | 1540.0000 | 1989.12.15 | 1993.04.15 | GNIP |
| BOGDARIN | 54.6200 | 113.1300 | 995.0000 | 1996.01.31 | 2000.12.31 | WaterIsotopes.org |
| BOLOGNA | 44.4833 | 11.3333 | 54.0000 | 1996.03.15 | 2001.08.31 | WaterIsotopes.org |
| BONGA | 13.0300 | 123.9200 | 600.0000 | 1997.01.15 | 1998.12.15 | GNIP |
| BOSSEA | 44.2333 | 7.8500 | 820.0000 | 2000.11.15 | 2001.03.15 | GNIP |
| BRAAKMAN | 51.3000 | 3.7500 | 2.0000 | 1988.07.15 | 1992.04.15 | GNIP |
| BRASIMONE | 44.0900 | 11.0800 | 842.0000 | 1971.09.15 | 1986.04.15 | GNIP |
| Brasimone | 44.1165 | 11.1157 | | 1971.09.01 | 1986.04.01 | References |

| | | | | | | |
|---|---|---|---|---|---|---|
| BRATISLAVA | 48.1691 | 17.1119 | 286.0000 | 1992.01.15 | 1992.02.15 | GNIP |
| BRAUNSCHWEIG | 52.2914 | 10.4464 | 81.0000 | 1990.01.15 | 2012.12.15 | GNIP |
| BRAVICEA | 47.4000 | 28.4917 | 78.0000 | 2007.09.15 | 2015.07.15 | GNIP |
| BREST | 52.0944 | 23.7058 | 136.0000 | 1981.01.15 | 1983.12.15 | GNIP |
| BREST PLOUZANE | 48.3600 | -4.5700 | 80.0000 | 1996.04.15 | 2002.12.15 | GNIP |
| BUCHS SUHR | 47.3723 | 8.0831 | 397.0000 | 1994.07.15 | 2019.12.15 | GNIP |
| BUSAN | 35.2306 | 129.0803 | 96.8000 | 2019.02.15 | 2020.06.15 | GNIP |
| CACERES | 39.4667 | -6.3333 | 405.0000 | 2000.03.15 | 2015.12.15 | GNIP |
| CAGBULACAO | 11.1200 | 124.6200 | 200.0000 | 1997.02.15 | 1998.12.15 | GNIP |
| CAGLIARI | 39.2167 | 9.1167 | 25.0000 | 1998.08.31 | 1998.08.31 | WaterIsotopes.org |
| CAHUL | 45.8000 | 28.2000 | 113.0000 | 2007.09.15 | 2015.08.15 | GNIP |
| CAIRO | 30.0800 | 31.2800 | 34.0000 | 1987.03.15 | 2003.03.15 | GNIP |
| CAMERON HIGHLANDS | 4.4667 | 101.3833 | 1430.0000 | 1996.02.15 | 2016.12.15 | GNIP |
| CAMPISTROUS | 43.1200 | 0.3800 | 600.0000 | 1997.10.15 | 1998.12.15 | GNIP |
| CAMPO CARLO MAGNO | 46.2333 | 10.8167 | 1685.0000 | 1999.09.30 | 1999.09.30 | WaterIsotopes.org |
| CANINO | 42.4667 | 11.7500 | 229.0000 | 1999.08.31 | 1999.08.31 | WaterIsotopes.org |
| CARPENTRAS | 44.9500 | 5.7800 | 99.0000 | 1997.02.15 | 1998.12.15 | GNIP |
| CASALE MONFERRATO | 45.1400 | 8.4500 | 116.0000 | 2002.02.28 | 2002.02.28 | WaterIsotopes.org |
| CBS | 48.0700 | 114.5200 | 759.0000 | 2002.10.01 | 2003.09.01 | WaterIsotopes.org |
| CEGLIE MESSAPICO | 40.6500 | 17.5167 | 302.0000 | 1999.10.31 | 1999.10.31 | WaterIsotopes.org |
| Centre for Earth Science, IIS | 13.0181 | 77.5693 | | 2010.06.07 | 2010.09.16 | WaterIsotopes.org |
| CESTAS-PIERROTON | 44.7381 | -0.7747 | 59.0000 | 2007.02.15 | 2018.12.15 | GNIP |
| CHANGCHUN | 43.9000 | 125.2167 | 237.0000 | 1999.05.15 | 2001.11.15 | GNIP |
| CHANGSHA | 28.2000 | 113.0667 | 37.0000 | 1988.01.15 | 1992.12.15 | GNIP |
| CHARCHES | 37.2932 | -2.9559 | 1426.0000 | 2013.01.15 | 2015.11.15 | GNIP |
| Chat-11 | 44.8000 | 34.2900 | | 2011.01.01 | 2011.12.01 | References |
| CHENGDU | 30.6700 | 104.0200 | 506.0000 | 1986.07.15 | 1998.05.15 | GNIP |
| CHEONGJU | 36.6200 | 127.4600 | 62.0000 | 1999.01.15 | 2016.12.15 | GNIP |
| CHERSKIY | 68.7594 | 161.3425 | 30.0000 | 2005.08.15 | 2010.06.15 | GNIP |
| Chiba | 35.3700 | 140.6000 | 20.0000 | 2013.02.01 | 2013.12.31 | WaterIsotopes.org |
| Chibi | 29.7200 | 113.8800 | | 2021.04.22 | 2022.02.22 | Measured data |

| | | | | | | |
|---|---|---|---|---|---|---|
| Chichibu | 35.5600 | 138.5100 | 800.0000 | 2013.01.31 | 2014.01.06 | WaterIsotopes.org |
| CHISINAU | 46.9667 | 28.9000 | 125.0000 | 2007.08.15 | 2015.07.15 | GNIP |
| CHONGQING (CUNTAN JIANG) | 29.6200 | 106.6000 | 192.0000 | 1992.06.15 | 1992.10.15 | GNIP |
| CHOPOK | 48.9330 | 19.5830 | 2008.0000 | 1991.08.15 | 1992.02.15 | GNIP |
| ChP-12 | 44.7556 | 34.3411 | | 2012.01.01 | 2015.12.08 | References |
| CHUADANGA | 23.6400 | 88.8500 | 20.0000 | 2014.05.15 | 2018.12.15 | GNIP |
| CIUDAD REAL | 38.9892 | -3.9203 | 628.0000 | 2000.01.15 | 2015.12.15 | GNIP |
| CKM | 36.5355 | 138.0619 | 763.0000 | 2011.08.15 | 2013.10.15 | WaterIsotopes.org |
| CLUJ | 46.7800 | 23.6100 | 330.0000 | 2015.01.15 | 2016.12.15 | GNIP |
| COMACCHIO | 44.6975 | 12.1858 | 1.0000 | 2001.05.31 | 2001.05.31 | WaterIsotopes.org |
| Cona Lake West | 32.0568 | 91.4201 | 4623.0000 | 2011.05.12 | 2013.10.21 | WaterIsotopes.org |
| Corchia | 44.0206 | 10.2842 | | 2009.10.01 | 2015.10.01 | References |
| COSENZA | 39.3000 | 16.2500 | 238.0000 | 2001.07.31 | 2001.07.31 | WaterIsotopes.org |
| Coxs Bazar | 21.4420 | 91.9690 | | 2015.03.05 | 2015.12.18 | WaterIsotopes.org |
| COX'S BAZAR | 21.4400 | 91.9900 | 5.0000 | 2014.05.15 | 2018.10.15 | GNIP |
| CUXHAVEN | 53.8713 | 8.7058 | 5.0000 | 1990.01.15 | 2012.12.15 | GNIP |
| DALBAHCE | 39.6667 | 43.8583 | 1740.0000 | 1990.04.15 | 1991.11.15 | GNIP |
| DAMASCUS | 33.4200 | 36.5200 | 609.0000 | 1989.12.15 | 1993.04.15 | GNIP |
| DARFO-BOARIO | 45.8833 | 10.1833 | 208.0000 | 1999.01.31 | 1999.01.31 | WaterIsotopes.org |
| Datanxiang | 38.4600 | 103.1400 | | 2020.07.24 | 2021.09.05 | Measured data |
| DAX | 43.6833 | -1.0667 | 9.0000 | 1999.01.15 | 2005.01.15 | GNIP |
| DE BILT | 52.1000 | 5.1800 | 2.0000 | 1981.01.15 | 1991.10.15 | GNIP |
| DE KOOY | 52.9300 | 4.7800 | 1.0000 | 1981.01.15 | 1987.12.15 | GNIP |
| DEVPRAYAG | 30.1406 | 78.5967 | 465.0000 | 2004.06.15 | 2006.10.15 | GNIP |
| DHAKA | 23.9528 | 90.2792 | 14.0000 | 2009.03.15 | 2018.12.15 | GNIP |
| Dilijan | 40.7419 | 44.8636 | 2104.2000 | 2014.07.01 | 2015.04.21 | WaterIsotopes.org |
| DILIMAN QUEZON CITY | 14.6400 | 121.0400 | 42.0000 | 2000.01.15 | 2016.12.15 | GNIP |
| DINAJPUR | 25.6200 | 88.6600 | 35.0000 | 2014.02.15 | 2018.12.15 | GNIP |
| DIYARBAKIR | 37.9011 | 40.2036 | 686.0000 | 2008.09.15 | 2016.12.15 | GNIP |
| DOBRANI | 30.9461 | 78.6881 | 2050.0000 | 2004.08.15 | 2006.09.15 | GNIP |
| DONG HOI | 17.4600 | 106.6200 | 1.0000 | 2014.01.15 | 2018.12.15 | GNIP |
| DRAIX | 44.1333 | 6.3333 | 851.0000 | 2004.02.15 | 2018.12.15 | GNIP |

| | | | | | |
|---|---|---|---|---|---|
| DRESDEN | 51.0500 | 13.7300 | 113.0000 | 1997.06.15 | 2001.01.15 | GNIP |
| DUBROVNIK | 42.6600 | 18.0833 | 52.0000 | 2000.09.15 | 2003.12.15 | GNIP |
| DUDINKA | 69.4075 | 86.1806 | 66.0000 | 1990.02.15 | 1990.12.15 | GNIP |
| DUGOPOLJE | 43.5833 | 16.6000 | 295.0000 | 2008.03.15 | 2009.01.15 | GNIP |
| Dunhuang | 40.1141 | 94.6028 | | 2021.05.13 | 2021.06.15 | Measured data |
| DURRAH | 29.3554 | 34.9603 | 0.0000 | 1968.11.15 | 1969.04.15 | GNIP |
| EDIRNE | 41.6781 | 26.5592 | 48.0000 | 2008.07.15 | 2016.12.15 | GNIP |
| EL-ARISH | 31.0800 | 33.8300 | 31.0000 | 2001.01.15 | 2003.03.15 | GNIP |
| EMMERICH | 51.8300 | 6.2500 | 43.0000 | 1990.01.15 | 2012.11.15 | GNIP |
| ENISEJSK | 58.4500 | 92.1500 | 78.0000 | 1990.01.15 | 1990.12.15 | GNIP |
| ERLANGEN (GEOZENTRUM) | 49.5971 | 11.0055 | 270.0000 | 2010.07.15 | 2018.12.15 | GNIP |
| ERZURUM | 39.9106 | 41.2756 | 1758.0000 | 2008.07.15 | 2016.12.15 | GNIP |
| ESPOO | 60.1801 | 24.8329 | 30.0000 | 2000.11.15 | 2017.12.15 | GNIP |
| Evaristo_SG | 1.2953 | 103.7711 | | 2014.11.11 | 2015.11.26 | WaterIsotopes.org |
| FANANO 1 | 44.2000 | 10.7833 | 1280.0000 | 1999.02.28 | 1999.02.28 | WaterIsotopes.org |
| FANANO 2 | 44.2000 | 10.7834 | 935.0000 | 1999.02.28 | 1999.02.28 | WaterIsotopes.org |
| FANANO 3 | 44.2000 | 10.7835 | 660.0000 | 1999.02.28 | 1999.02.28 | WaterIsotopes.org |
| FANO | 43.8333 | 13.0167 | 12.0000 | 1999.12.31 | 1999.12.31 | WaterIsotopes.org |
| Fantan | 40.3956 | 44.6681 | 1779.0000 | 2014.07.01 | 2015.02.02 | WaterIsotopes.org |
| FEHMARN | 54.5283 | 11.0603 | 3.0000 | 1997.06.15 | 2013.12.15 | GNIP |
| Fengjie | 31.0183 | 109.4648 | | 2022.01.25 | 2022.02.08 | Measured data |
| FIRENZE | 43.7667 | 11.2500 | 50.0000 | 2001.06.30 | 2001.06.30 | WaterIsotopes.org |
| FLEAM DYKE | 52.1667 | 0.2500 | 30.0000 | 1980.01.15 | 1983.12.15 | GNIP |
| FONTAINEBLEAU | 48.4042 | 2.6965 | 79.0000 | 2016.04.15 | 2018.09.15 | GNIP |
| Fukui | 36.3000 | 136.1300 | 10.0000 | 2013.02.01 | 2014.01.04 | WaterIsotopes.org |
| Funabashi | 35.4300 | 140.1000 | 30.0000 | 2013.02.01 | 2013.12.27 | WaterIsotopes.org |
| FUNCHAL | 32.6333 | -16.9000 | 58.0000 | 1990.01.15 | 2017.02.15 | GNIP |
| Furano | 43.1300 | 142.3500 | 400.0000 | 2013.01.29 | 2013.12.31 | WaterIsotopes.org |
| FUZHOU | 26.0833 | 119.2833 | 16.0000 | 1985.10.15 | 1992.12.15 | GNIP |

| | | | | | | |
|---|---|---|---|---|---|---|
| Fuzhou | 26.0277 | 119.2103 | | 2021.12.17 | 2022.04.20 | Measured data |
| GALLO | 41.3000 | 13.9833 | 825.0000 | 1998.09.30 | 1998.09.30 | WaterIsotopes.org |
| GANGOTRI | 30.9967 | 78.9403 | 3053.0000 | 2004.06.15 | 2006.09.15 | GNIP |
| Ganzi | 31.9800 | 100.0000 | | 2022.01.27 | 2022.05.01 | Measured data |
| Gaotai | 39.3783 | 99.8192 | | 2020.11.21 | 2022.02.18 | Measured data |
| GARDANNE | 43.4500 | 5.4500 | 215.0000 | 1997.04.15 | 1998.12.15 | GNIP |
| GARMISCH-PARTENKIRCHEN | 47.4828 | 11.0622 | 719.0000 | 1990.01.15 | 2013.12.15 | GNIP |
| GENOA | 44.4200 | 8.8500 | 2.0000 | 1990.01.15 | 2002.01.15 | GNIP |
| GENOVA-SESTRI | 44.4167 | 8.9500 | 2.0000 | 1997.12.31 | 1997.12.31 | WaterIsotopes.org |
| GILZE-RIJEN | 51.5700 | 4.9300 | 11.0000 | 1981.01.15 | 1988.03.15 | GNIP |
| GIRONA | 41.9117 | 2.7633 | 129.0000 | 2000.03.15 | 2015.12.15 | GNIP |
| GOERLITZ | 51.1622 | 14.9506 | 238.0000 | 1997.06.15 | 2013.12.15 | GNIP |
| Gokase | 32.4100 | 131.1100 | 550.0000 | 2013.02.01 | 2013.12.20 | WaterIsotopes.org |
| GOMUKH | 30.9261 | 78.9403 | 3800.0000 | 2004.05.15 | 2006.09.15 | GNIP |
| GOR'KIJ | 56.2167 | 43.8167 | 82.0000 | 1981.01.15 | 1983.12.15 | GNIP |
| GOSPIC | 44.5333 | 15.3833 | 564.0000 | 2008.02.15 | 2008.12.15 | GNIP |
| GRANIGA | 46.0700 | 8.1100 | 1100.0000 | 2000.12.31 | 2000.12.31 | WaterIsotopes.org |
| GRAZ | 47.0778 | 15.4489 | 366.0000 | 1990.01.15 | 2002.12.15 | GNIP |
| GREIFSWALD | 54.0967 | 13.4056 | 2.0000 | 2002.07.15 | 2013.12.15 | GNIP |
| GRIMSEL | 46.5725 | 8.3327 | 1950.0000 | 1971.01.15 | 2019.12.15 | GNIP |
| GRONINGEN | 53.2300 | 6.5500 | 1.0000 | 1990.01.15 | 2012.12.15 | GNIP |
| Guangyuan | 32.4337 | 105.8300 | | 2021.12.10 | 2022.05.09 | Measured data |
| GUANGZHOU | 23.1300 | 113.3200 | 7.0000 | 2007.01.11 | 2009.11.14 | WaterIsotopes.org |
| Guangzhou | 23.1600 | 113.2300 | | 2021.12.19 | 2022.02.22 | Measured data |
| GUILIN | 25.0700 | 110.0800 | 170.0000 | 1983.01.15 | 1990.12.15 | GNIP |
| GUIYANG | 26.5833 | 106.7167 | 1071.0000 | 1988.02.15 | 1992.12.15 | GNIP |
| Guizhou | 26.8000 | 106.7800 | | 2022.01.04 | 2022.05.11 | Measured data |
| GUTTANNEN | 46.6577 | 8.2927 | 1055.0000 | 2002.01.15 | 2019.12.15 | GNIP |
| GUWAHATI | 26.1908 | 91.7953 | 54.0000 | 2003.07.15 | 2004.10.15 | GNIP |
| Guyuan | 36.0100 | 106.2800 | | 2020.11.20 | 2022.02.16 | Measured data |

| | | | | | | |
|---|---|---|---|---|---|---|
| H-4 RWASHED | 32.5025 | 38.1945 | 686.0000 | 1968.11.15 | 1969.04.15 | GNIP |
| H-5 SAFAWI | 32.1991 | 37.1026 | 715.0000 | 1968.12.15 | 1969.04.15 | GNIP |
| HABAROVSK | 48.5200 | 135.1700 | 72.0000 | 1971.01.15 | 1971.12.15 | GNIP |
| HAERBIN | 45.6800 | 126.6200 | 172.0000 | 1986.06.15 | 1997.08.15 | GNIP |
| Hagi | 34.2400 | 131.2300 | 5.0000 | 2013.02.01 | 2014.01.04 | WaterIsotopes.org |
| HAIKOU | 20.0333 | 110.3500 | 15.0000 | 1988.03.15 | 2000.07.15 | GNIP |
| HAMBANTOTA | 6.1167 | 81.1333 | 20.0000 | 1983.01.15 | 1986.06.15 | GNIP |
| HANOI | 21.0453 | 105.7987 | 11.0000 | 2004.09.15 | 2007.12.15 | GNIP |
| HANOI (IGS) | 21.0300 | 105.8400 | 15.0000 | 2015.01.15 | 2018.12.15 | GNIP |
| Hebei Township | 34.7000 | 100.7857 | | 2021.07.23 | 2021.09.19 | Measured data |
| Heidelberg | 49.8000 | 8.7000 | 110.0000 | 1981.01.31 | 1988.06.30 | WaterIsotopes.org |
| Hemuqiao watershed | 34.3100 | 119.3400 | | 2017.10.01 | 2020.05.31 | References |
| HETIAN | 37.1333 | 79.9333 | 1375.0000 | 1988.02.15 | 1992.12.15 | GNIP |
| Hezuo | 35.0000 | 102.9100 | | 2021.03.26 | 2021.11.11 | Measured data |
| Higashihiroshima | 34.2300 | 132.4200 | 220.0000 | 2013.02.01 | 2013.08.01 | WaterIsotopes.org |
| Hikone | 35.1500 | 136.1200 | 90.0000 | 2013.02.01 | 2014.01.01 | WaterIsotopes.org |
| Ho Chi Minh City | 10.0430 | 106.6880 | | 2013.05.15 | 2015.12.21 | WaterIsotopes.org |
| HOF-HOHENSAAS | 50.3119 | 11.8758 | 565.0000 | 1990.01.15 | 2013.12.15 | GNIP |
| Hofu | 34.2000 | 131.3400 | 5.0000 | 2013.02.01 | 2013.12.30 | WaterIsotopes.org |
| HOHENPEISSENBERG | 47.8008 | 11.0108 | 977.0000 | 1990.01.15 | 2008.10.15 | GNIP |
| HOMS | 34.7500 | 36.7167 | 490.0000 | 1989.12.15 | 1993.05.15 | GNIP |
| HONG KONG | 22.3167 | 114.1667 | 66.0000 | 1990.01.15 | 2018.12.15 | GNIP |
| Hongqigu | 38.2100 | 102.5000 | | 2019.07.09 | 2021.09.06 | Measured data |
| HONGSEONG | 36.5553 | 126.6359 | 62.0000 | 2018.03.15 | 2020.08.15 | GNIP |
| Hongya County | 29.9066 | 103.3730 | | 2020.07.22 | 2022.05.09 | Measured data |
| Huangshuigou | 37.6900 | 101.9100 | | 2021.11.27 | 2021.11.27 | Measured data |
| Huanjiang County | 24.4300 | 108.1800 | | 2011.06.01 | 2020.11.01 | References |
| Hulinzhan | 37.4100 | 101.5300 | | 2019.08.19 | 2021.11.27 | Measured data |

| | | | | | |
|---|---|---|---|---|---|
| HYDERABAD | 17.4500 | 78.4700 | 545.0000 | 1997.09.15 | 2001.04.15 | GNIP |
| Hynek_04-AB | 40.7544 | 75.1867 | 3508.0000 | 2004.07.15 | 2004.07.17 | WaterIsotopes.org |
| Hynek_04-CTS | 41.7742 | 78.3999 | 4457.0000 | 2004.07.29 | 2004.07.31 | WaterIsotopes.org |
| Hynek_04-IK | 42.6595 | 77.2028 | 1620.3333 | 2004.04.30 | 2004.08.17 | WaterIsotopes.org |
| Hynek_04-WA | 35.8232 | 74.6504 | 1426.4640 | 2004.07.01 | 2004.07.01 | WaterIsotopes.org |
| Hynek_05-AB | 40.7441 | 75.2155 | 3760.0000 | 2005.09.06 | 2005.09.06 | WaterIsotopes.org |
| Hynek_05-WA | 36.8500 | 75.4281 | 4719.5232 | 2004.07.01 | 2004.07.01 | WaterIsotopes.org |
| Hynek_06-WA | 41.4587 | 76.4511 | 2253.0816 | 2004.07.01 | 2004.07.01 | WaterIsotopes.org |
| Hynek_07-WA | 41.4587 | 76.4511 | 2253.0816 | 2004.07.01 | 2004.07.01 | WaterIsotopes.org |
| Hynek_08-WA | 41.7568 | 75.1089 | 2567.0256 | 2004.07.01 | 2004.07.01 | WaterIsotopes.org |
| Hynek_09-WA | 41.7643 | 75.2801 | 3054.4008 | 2004.07.01 | 2004.07.01 | WaterIsotopes.org |
| Hynek_10-WA | 40.8231 | 75.2893 | 3031.0000 | 2004.07.01 | 2004.07.01 | WaterIsotopes.org |
| Hynek_Bos-Prec | 42.6587 | 77.2031 | 1644.2500 | 2004.03.11 | 2004.05.30 | WaterIsotopes.org |
| Hynek_IK | 42.6606 | 77.2026 | 1635.0000 | 2003.07.05 | 2003.11.25 | WaterIsotopes.org |
| Hynek_IL | 42.6452 | 74.2751 | 1581.0000 | 2008.09.03 | 2008.09.03 | WaterIsotopes.org |
| Hynek_KB | 42.9391 | 78.3258 | 2258.0000 | 2003.06.28 | 2003.06.28 | WaterIsotopes.org |
| Hynek_KYR07 | 40.8194 | 75.2281 | 3010.0000 | 2007.09.05 | 2007.09.05 | WaterIsotopes.org |
| Hynek_TAJ07 | 37.6453 | 74.3535 | 4100.0000 | 2007.09.05 | 2007.09.08 | WaterIsotopes.org |
| Ichinomiya | 35.2200 | 140.2200 | 5.0000 | 2013.02.02 | 2013.10.01 | WaterIsotopes.org |
| IDLEB | 35.9392 | 36.6067 | 451.0000 | 1992.12.15 | 1993.05.15 | GNIP |
| IKW | 35.2231 | 138.2231 | 755.0000 | 2011.05.15 | 2015.06.15 | WaterIsotopes.org |
| Imizu | 36.4200 | 137.5000 | 20.0000 | 2013.01.31 | 2013.12.31 | WaterIsotopes.org |
| IMPERIA | 43.8833 | 8.0500 | 10.0000 | 2001.03.31 | 2001.03.31 | WaterIsotope |

| | | | | | |
|---|---|---|---|---|---|
| | | | | | s.org |
| INCHNADAMPH | 58.1500 | -4.9750 | 73.0000 | 2003.12.15 | 2005.10.15 | GNIP |
| IRKUTSK | 52.2700 | 104.3500 | 485.0000 | 1990.01.15 | 1990.11.15 | GNIP |
| Irkutsk | 52.3000 | 104.2833 | 469.0000 | 2011.06.24 | 2017.04.04 | WaterIsotope s.org |
| Isesaki | 36.1500 | 139.1200 | 50.0000 | 2013.02.01 | 2013.12.27 | WaterIsotope s.org |
| ISFJORD RADIO | 78.0700 | 13.6300 | 6.0000 | 1961.07.15 | 1975.05.15 | GNIP |
| ISO_NS_Birgit | 47.7347 | 12.8843 | 470.0000 | 2012.06.18 | 2012.11.14 | WaterIsotope s.org |
| ISO_NS_GC | 47.7885 | 12.9803 | 442.0000 | 2012.05.24 | 2013.02.01 | WaterIsotope s.org |
| ISO_NS_Rothmann bach | 47.7035 | 13.0363 | 465.0000 | 2012.06.23 | 2012.12.05 | WaterIsotope s.org |
| ISO_NS_Salzburger Hochthron | 47.7165 | 13.0053 | 1829.0000 | 2012.05.25 | 2012.10.10 | WaterIsotope s.org |
| ISO_NS_Veitlbruch | 47.7332 | 12.9738 | 639.0000 | 2012.05.24 | 2012.12.04 | WaterIsotope s.org |
| ISO_NS_Zeppezaue r Haus | 47.7245 | 13.0059 | 1649.0000 | 2012.05.25 | 2012.10.10 | WaterIsotope s.org |
| IZMIR | 38.4306 | 27.1511 | 120.0000 | 2008.09.15 | 2016.12.15 | GNIP |
| IZRAA | 32.8389 | 36.2569 | 580.0000 | 1989.12.15 | 1993.04.15 | GNIP |
| Izumi | 32.5000 | 130.2100 | 15.0000 | 2013.02.01 | 2013.12.21 | WaterIsotope s.org |
| JAFFNA | 9.6678 | 80.0065 | 6.0000 | 2014.01.15 | 2018.12.15 | GNIP |
| JAKARTA | -6.1800 | 106.8300 | 8.0000 | 1980.05.15 | 1997.12.15 | GNIP |
| JAMMU | 32.6925 | 74.8461 | 367.0000 | 2003.07.15 | 2006.09.15 | GNIP |
| JARABLOUS | 36.8222 | 38.0125 | 351.0000 | 1991.12.15 | 1993.05.15 | GNIP |
| JAYAPURA | -2.5300 | 140.7200 | 3.0000 | 1980.01.15 | 1991.06.15 | GNIP |
| Jingtai | 37.1836 | 104.0630 | | 2020.08.04 | 2021.08.05 | Measured data |
| Jinta County | 39.9843 | 98.9032 | | 2021.05.14 | 2022.02.17 | Measured data |
| Jiudun | 38.0700 | 102.4500 | | 2019.07.13 | 2021.10.08 | Measured data |
| Jiuzhaigou | 33.2632 | 104.2367 | | 2020.07.23 | 2021.10.18 | Measured data |
| JOHOR BAHRU | 1.6000 | 103.6667 | 35.0000 | 2002.03.15 | 2016.12.15 | GNIP |
| KABUL | 34.5700 | 69.2133 | 1860.0000 | 1970.01.15 | 1989.09.15 | GNIP |
| Kagoshima | 31.3700 | 130.3100 | 15.0000 | 2013.03.31 | 2013.07.31 | WaterIsotope s.org |
| Kakamigahara | 35.2400 | 136.5200 | 40.0000 | 2013.02.01 | 2014.01.01 | WaterIsotope s.org |

| | | | | | |
|---|---|---|---|---|---|
| KAKINADA | 17.0211 | 82.2567 | 8.0000 | 2003.07.15 | 2006.09.15 | GNIP |
| KALININ | 56.9000 | 35.9000 | 31.0000 | 1981.01.15 | 1987.07.15 | GNIP |
| KANDALAKSA | 67.1500 | 32.3500 | 26.0000 | 1996.01.15 | 2000.12.15 | GNIP |
| Kapan | 39.2047 | 46.4461 | 704.0000 | 2014.07.07 | 2015.05.06 | WaterIsotopes.org |
| KARACHI | 24.9000 | 67.1300 | 23.0000 | 1961.07.15 | 1973.12.15 | GNIP |
| KARLSRUHE | 49.0392 | 8.3650 | 112.0000 | 1990.01.15 | 2013.12.15 | GNIP |
| Kashgar | 39.4704 | 75.9898 | | 2020.08.29 | 2021.06.16 | Measured data |
| KBU | 47.2800 | 108.7800 | 1255.0000 | 2002.10.01 | 2003.09.01 | WaterIsotopes.org |
| KEYWORTH | 52.8833 | -1.0833 | 60.0000 | 1985.01.15 | 1996.12.15 | GNIP |
| KHANTY-MANSIYSK | 60.9667 | 69.0667 | 40.0000 | 1996.01.15 | 1997.03.15 | GNIP |
| KHARKIV | 49.9333 | 36.2833 | 148.0000 | 2013.11.15 | 2016.12.15 | GNIP |
| Kinko | 31.1400 | 130.4700 | 20.0000 | 2013.02.01 | 2014.01.01 | WaterIsotopes.org |
| KIROV | 58.6500 | 49.6167 | 164.0000 | 1981.01.15 | 2000.12.15 | GNIP |
| Kiryu | 36.2400 | 139.1900 | 140.0000 | 2013.02.01 | 2014.01.01 | WaterIsotopes.org |
| KKC | 36.2533 | 137.6686 | 1529.0000 | 2010.07.15 | 2015.06.15 | WaterIsotopes.org |
| KLAGENFURT FLUGHAFEN | 46.6483 | 14.3183 | 450.0000 | 1990.01.15 | 2002.12.15 | GNIP |
| KO SAMUI | 9.4667 | 100.0500 | 7.0000 | 1980.01.15 | 1983.11.15 | GNIP |
| KO SICHANG | 13.1700 | 100.8000 | 26.0000 | 1984.06.15 | 1991.10.15 | GNIP |
| Kobayashi | 31.5900 | 130.5800 | 209.0000 | 2013.02.01 | 2013.12.22 | WaterIsotopes.org |
| Kobayashi2 | 31.5900 | 130.5800 | 668.0000 | 2013.02.01 | 2013.12.22 | WaterIsotopes.org |
| KOBLENZ | 50.3381 | 7.6000 | 85.0000 | 1990.01.15 | 2013.12.15 | GNIP |
| Kochi | 33.3200 | 133.2900 | 30.0000 | 2013.02.01 | 2014.01.06 | WaterIsotopes.org |
| KOF | 35.6828 | 138.5664 | 304.0000 | 2010.07.15 | 2015.06.15 | WaterIsotopes.org |
| KOLKATA | 22.7978 | 88.3717 | 6.0000 | 2004.07.15 | 2006.10.15 | GNIP |
| KOMIZA-VIS ISLAND | 43.0392 | 16.0906 | 6.0000 | 2000.09.15 | 2003.12.15 | GNIP |
| KONSTANZ | 47.6772 | 9.1900 | 443.0000 | 1990.01.15 | 2013.12.15 | GNIP |
| KOTA BAHRU | 6.1667 | 102.2833 | 7.0000 | 1991.01.15 | 2016.12.15 | GNIP |
| KOTA KINABALU | 5.9300 | 116.0600 | 9.0000 | 2013.04.15 | 2016.12.15 | GNIP |
| KOZHIKODE | 11.2500 | 75.7800 | 20.0000 | 1997.05.15 | 2007.11.15 | GNIP |
| KOZINA | 45.6000 | 13.9333 | 497.0000 | 2000.10.15 | 2003.12.15 | GNIP |

| | | | | | | |
|---|---|---|---|---|---|---|
| K-puszta | 46.9600 | 19.5500 | | 2013.04.02 | 2017.12.15 | References |
| KRAKOW | 50.0617 | 19.8486 | 205.0000 | 1990.01.15 | 2016.12.15 | GNIP |
| KRENKEL POLAR GMO | 80.6167 | 58.0500 | 20.0000 | 1990.02.15 | 1990.12.15 | GNIP |
| KUALA LUMPUR | 2.8833 | 101.7833 | 26.0000 | 2000.01.15 | 2016.12.15 | GNIP |
| KUALA TERENGGANU | 5.3833 | 103.1000 | 10.0000 | 2001.05.15 | 2016.12.15 | GNIP |
| Kuala Terengganu | 5.4120 | 103.0850 | | 2014.12.04 | 2016.09.12 | WaterIsotopes.org |
| KUANTAN | 3.7833 | 103.3333 | 1.0000 | 2013.04.15 | 2016.12.15 | GNIP |
| KUCHING | 1.5300 | 110.3400 | 15.0000 | 2013.05.15 | 2016.12.15 | GNIP |
| Kuching | 1.4590 | 119.4130 | | 2014.07.30 | 2016.02.29 | WaterIsotopes.org |
| Kuju | 33.1000 | 131.1700 | 550.0000 | 2013.02.01 | 2014.01.01 | WaterIsotopes.org |
| KUMAMOTO | 32.8133 | 130.7292 | 26.0000 | 2015.04.15 | 2015.12.15 | GNIP |
| Kumamoto | 32.4800 | 130.4300 | 30.0000 | 2013.01.31 | 2013.12.28 | WaterIsotopes.org |
| KUNGUR | 57.4368 | 56.9593 | 120.0000 | 2016.12.15 | 2018.06.15 | GNIP |
| KUNMING | 25.0167 | 102.6833 | 1892.0000 | 1988.02.15 | 2003.12.15 | GNIP |
| KUOPIO | 62.8918 | 27.6254 | 116.0000 | 2005.01.15 | 2017.12.15 | GNIP |
| KURSK | 51.7700 | 36.1700 | 247.0000 | 1996.01.15 | 2000.12.31 | WaterIsotopes.org |
| Kushimoto | 33.2600 | 135.4500 | 50.0000 | 2013.01.31 | 2014.01.03 | WaterIsotopes.org |
| Kyoto | 35.4000 | 135.4500 | 140.0000 | 2013.02.01 | 2014.01.06 | WaterIsotopes.org |
| Kyuji County | 33.4299 | 101.4830 | | 2021.03.26 | 2021.10.20 | Measured data |
| LA BREVINE | 46.9814 | 6.6080 | 1042.0000 | 1994.01.15 | 2019.12.15 | GNIP |
| Lake Massaciuccioli | 43.8376 | 10.3524 | | 2007.03.01 | 2014.03.01 | References |
| lake Shortandy | 52.9889 | 70.2185 | 390.0000 | 2015.12.22 | 2016.11.02 | WaterIsotopes.org |
| LANZHOU | 36.0500 | 103.8800 | 1517.0000 | 1985.07.15 | 1999.07.01 | GNIP |
| L'AQUILA | 42.3667 | 13.3667 | 710.0000 | 2001.04.30 | 2001.04.30 | WaterIsotopes.org |
| LEIPZIG | 51.3500 | 12.4300 | 125.0000 | 1990.01.15 | 2013.12.15 | GNIP |
| LEON VIRGEN DEL CAMINO | 42.5883 | -5.6511 | 916.0000 | 2000.02.15 | 2015.12.15 | GNIP |
| LEOVA | 46.4972 | 28.3000 | 156.0000 | 2007.09.15 | 2015.08.15 | GNIP |
| Leshan | 29.5820 | 103.7610 | | 2021.12.10 | 2022.05.26 | Measured data |
| LHASA | 29.7000 | 91.1333 | 3649.0000 | 1987.01.15 | 1992.12.15 | GNIP |

| | | | | | | |
|---|---|---|---|---|---|---|
| LICKO LESCE | 44.7833 | 15.3167 | 463.0000 | 2008.02.15 | 2008.12.15 | GNIP |
| LIEGE | 50.7000 | 5.4700 | 190.0000 | 1966.01.15 | 1970.11.15 | GNIP |
| LIESEK | 49.3600 | 19.6800 | 692.0000 | 1991.08.15 | 1992.02.15 | GNIP |
| Lijiaxia | 39.9042 | 116.4000 | | 2020.06.21 | 2020.09.08 | Measured data |
| LIPTOVSKY MIKULAS-ONDR ASOVA | 49.0975 | 19.5900 | 570.0000 | 1992.01.15 | 2016.12.15 | GNIP |
| LISBON-ITN | 38.7927 | -9.1057 | 12.0000 | 2003.01.15 | 2017.05.15 | GNIP |
| LISTA | 58.1000 | 6.5700 | 13.0000 | 1961.02.15 | 1977.08.15 | GNIP |
| Liujiaxia | 33.1900 | 106.5700 | | 2020.06.21 | 2021.10.19 | Measured data |
| LIUZHOU | 24.3500 | 109.4000 | 97.0000 | 1988.01.15 | 1992.08.15 | GNIP |
| Lixian | 34.2200 | 105.1500 | | 2020.12.13 | 2021.11.29 | Measured data |
| LJUBLJANA | 46.0950 | 14.5970 | 282.0000 | 1990.01.15 | 2010.12.15 | GNIP |
| LOCARNO | 46.1738 | 8.7886 | 379.0000 | 1990.10.15 | 2019.12.15 | GNIP |
| Loch a Chem Alltain | 58.6569 | -4.9362 | 90.0000 | 2015.03.31 | 2015.03.31 | WaterIsotope s.org |
| Loch Bad an Losguinn | 57.0882 | -5.0415 | 240.0000 | 2015.03.31 | 2015.03.31 | WaterIsotope s.org |
| Loch Bealach Cornaidh | 58.2061 | -5.0515 | 428.0000 | 2015.03.31 | 2015.03.31 | WaterIsotope s.org |
| Loch Clair | 57.5623 | -5.3469 | 93.0000 | 2015.03.31 | 2015.03.31 | WaterIsotope s.org |
| Loch Coire Fionnaraich | 57.4917 | -5.4306 | 238.0000 | 2015.03.31 | 2015.03.31 | WaterIsotope s.org |
| Loch Doilean | 56.7501 | -5.5868 | 10.0000 | 2015.03.31 | 2015.03.31 | WaterIsotope s.org |
| Loch Dubh Cadhafuaraich | 58.1339 | -4.2400 | 400.0000 | 2015.03.31 | 2015.03.31 | WaterIsotope s.org |
| Loch Dubh Camas an Lochain | 57.9132 | -5.5957 | 24.0000 | 2015.03.31 | 2015.03.31 | WaterIsotope s.org |
| Loch Laidon | 56.6515 | -4.6440 | 282.0000 | 2015.03.31 | 2015.03.31 | WaterIsotope s.org |
| Loch na Achlaise | 56.5934 | -4.7540 | 294.0000 | 2015.03.31 | 2015.03.31 | WaterIsotope s.org |
| Loch na Creige Duibhe | 58.0503 | -5.3823 | 100.0000 | 2015.03.31 | 2015.03.31 | WaterIsotope s.org |
| Loch Nan Eion | 57.4997 | -5.4647 | 352.0000 | 2015.03.31 | 2015.03.31 | WaterIsotope s.org |
| Loch nan Eun | 58.2214 | -5.0119 | 164.0000 | 2015.03.31 | 2015.03.31 | WaterIsotope s.org |

| | | | | | |
|---|---|---|---|---|---|
| Loch Tinker | 56.2283 | -4.5099 | 420.0000 | 2015.03.31 | 2015.03.31 | WaterIsotopes.org |
| Loch Toll an Lochain | 57.7969 | -5.2424 | 517.0000 | 2015.03.31 | 2015.03.31 | WaterIsotopes.org |
| Lochan an Dubha | 58.0000 | -5.1375 | 84.0000 | 2015.03.31 | 2015.03.31 | WaterIsotopes.org |
| Lochan Dubh | 56.7828 | -5.4475 | 232.0000 | 2015.03.31 | 2015.03.31 | WaterIsotopes.org |
| Lochan Feoir | 58.1800 | -5.0135 | 117.0000 | 2015.03.31 | 2015.03.31 | WaterIsotopes.org |
| Lochan Fhionnlaidh | 58.0448 | -5.0665 | 230.0000 | 2015.03.31 | 2015.03.31 | WaterIsotopes.org |
| Lochan Lairig Cheile | 56.4203 | -4.3393 | 290.0000 | 2015.03.31 | 2015.03.31 | WaterIsotopes.org |
| Lochnagar | 56.9584 | -3.2333 | 785.0000 | 2015.03.31 | 2015.03.31 | WaterIsotopes.org |
| Longyangxia | 35.9870 | 100.7096 | | 2021.03.08 | 2021.10.26 | Measured data |
| LUANG-PRABANG | 19.8800 | 102.1300 | 305.0000 | 1961.03.15 | 1964.11.15 | GNIP |
| LUCERA | 41.5000 | 15.3333 | 219.0000 | 1998.11.30 | 1998.11.30 | WaterIsotopes.org |
| L'VOV | 49.8167 | 23.9500 | 329.0000 | 1981.01.15 | 1983.12.15 | GNIP |
| Mado | 34.9200 | 98.2091 | | 2021.03.31 | 2021.11.01 | Measured data |
| MADRID-RETIRO | 40.4120 | -3.6781 | 667.0000 | 1990.01.15 | 2015.12.15 | GNIP |
| MALAUSSENE | 43.9200 | 7.1300 | 359.0000 | 1997.06.15 | 1998.09.15 | GNIP |
| MALINSKA-KRK ISLAND | 45.1208 | 14.5261 | 1.0000 | 2000.12.15 | 2001.11.15 | GNIP |
| Mamurogawa | 38.5600 | 140.1500 | 180.0000 | 2013.01.30 | 2014.01.08 | WaterIsotopes.org |
| Manshuitan | 37.4900 | 104.0100 | | 2020.08.04 | 2020.09.21 | Measured data |
| MANTOVA | 45.1500 | 10.8000 | 19.0000 | 2001.01.31 | 2001.01.31 | WaterIsotopes.org |
| Maqin | 34.4775 | 100.2390 | | 2021.03.03 | 2021.10.28 | Measured data |
| Maqu | 33.9987 | 102.0727 | | 2021.04.06 | 2021.09.04 | Measured data |
| MARINA DI RAGUSA | 36.8330 | 14.5500 | 5.0000 | 2001.01.31 | 2001.01.31 | WaterIsotopes.org |
| Matsue | 35.2900 | 133.5000 | 10.0000 | 2013.02.01 | 2014.01.02 | WaterIsotopes.org |

| | | | | | | |
|---|---|---|---|---|---|---|
| MDG | 45.7500 | 106.2700 | 1393.0000 | 2002.10.01 | 2003.08.01 | WaterIsotopes.org |
| Meguro | 35.3900 | 139.4000 | 40.0000 | 2013.02.01 | 2014.01.01 | WaterIsotopes.org |
| Meikuang | 37.3300 | 101.5100 | | 2019.05.04 | 2021.11.27 | Measured data |
| MEIRINGEN | 46.7274 | 8.1868 | 632.0000 | 1990.01.15 | 2019.12.15 | GNIP |
| MILANO | 45.4667 | 9.2000 | 122.0000 | 2001.02.28 | 2001.02.28 | WaterIsotopes.org |
| MILHOSTOV | 48.6583 | 21.7300 | 104.0000 | 1991.08.15 | 1991.12.15 | GNIP |
| Minamiizu | 34.3800 | 138.5100 | 30.0000 | 2013.02.04 | 2014.01.06 | WaterIsotopes.org |
| MINSK | 53.9300 | 27.6300 | 225.0000 | 1981.01.15 | 1983.12.15 | GNIP |
| Miyake | 34.4000 | 139.2900 | 50.0000 | 2013.01.31 | 2013.05.28 | WaterIsotopes.org |
| Miyakonojo | 31.4400 | 131.3000 | 149.0000 | 2013.01.31 | 2013.12.31 | WaterIsotopes.org |
| Miyazaki | 31.5400 | 131.2400 | 10.0000 | 2013.02.02 | 2014.01.06 | WaterIsotopes.org |
| MNG | 48.2000 | 108.5000 | 1439.0000 | 2002.10.01 | 2003.09.01 | WaterIsotopes.org |
| MOCHOVCE | 48.2846 | 18.4757 | 206.0000 | 1991.08.15 | 1992.02.15 | GNIP |
| MODENA | 44.6667 | 10.9167 | 34.0000 | 2001.09.30 | 2001.09.30 | WaterIsotopes.org |
| Molinos | 40.7900 | -0.4500 | | 2010.03.02 | 2012.08.29 | WaterIsotopes.org |
| MONACO | 43.7324 | 7.4236 | 2.0000 | 1999.07.15 | 2016.12.15 | GNIP |
| Monatssammelprobe Station Baden Stadtgartenamt | 48.0113 | 16.2348 | | 1985.07.01 | 1986.02.01 | WaterIsotopes.org |
| MONTE CONERO | 43.5498 | 13.6012 | 530.0000 | 2014.01.15 | 2018.12.15 | GNIP |
| MONTPELLIER | 43.5700 | 3.9500 | 45.0000 | 1997.04.15 | 1998.12.15 | GNIP |
| Morioka | 39.4600 | 141.7000 | 193.0000 | 2013.01.31 | 2014.01.06 | WaterIsotopes.org |
| MORON DE LA FRONTERA | 37.1645 | -5.6114 | 87.0000 | 2000.03.15 | 2015.12.15 | GNIP |
| MOSCOW | 55.7500 | 37.5700 | 157.0000 | 1970.01.15 | 1979.11.15 | GNIP |
| MTM | 36.2511 | 137.9778 | 620.0000 | 2010.07.15 | 2015.06.15 | WaterIsotopes.org |
| Mulu | 4.0500 | 114.8100 | | 2013.01.01 | 2017.05.08 | WaterIsotopes.org |
| Mulu Airport | 4.0500 | 114.8100 | 24.0000 | 2006.07.03 | 2012.12.04 | WaterIsotopes.org |

| | | | | | |
|---|---|---|---|---|---|
| MURCIA | 38.0019 | -1.1708 | 61.0000 | 2000.03.15 | 2015.12.15 | GNIP |
| MURMANSK | 68.9667 | 33.0500 | 46.0000 | 1981.01.15 | 1990.03.15 | GNIP |
| Mutouqiao | 37.4800 | 101.5900 | | 2021.11.27 | 2021.11.27 | Measured data |
| Nagasaki | 32.4700 | 129.5100 | 20.0000 | 2013.01.31 | 2014.01.06 | WaterIsotopes.org |
| Nagoya | 35.1500 | 136.9700 | | 2013.06.12 | 2017.05.27 | WaterIsotopes.org |
| NAIMAKKA | 68.6833 | 21.5300 | 403.0000 | 1990.01.15 | 1995.12.15 | GNIP |
| NANJING | 32.1800 | 118.1800 | 26.0000 | 1988.01.15 | 1992.12.15 | GNIP |
| Nanyang Technological University | 1.3460 | 103.6790 | | 2013.11.28 | 2016.10.03 | WaterIsotopes.org |
| Nara | 34.4000 | 135.5000 | 115.0000 | 2013.02.01 | 2014.01.06 | WaterIsotopes.org |
| NBY | 35.9452 | 138.4697 | 1350.0000 | 2010.07.15 | 2015.06.15 | WaterIsotopes.org |
| NEUBRANDENBURG | 53.5483 | 13.1931 | 81.0000 | 1997.06.15 | 2002.06.15 | GNIP |
| NEUHERBERG | 48.2200 | 11.5900 | 489.0000 | 1988.01.15 | 2001.12.15 | GNIP |
| NEW DELHI | 28.5800 | 77.2000 | 212.0000 | 1980.01.15 | 2007.09.15 | GNIP |
| NGN | 36.6317 | 138.1891 | 354.0000 | 2010.07.15 | 2015.06.15 | WaterIsotopes.org |
| Niederschlagsstation Vordernbachalm | 47.6604 | 13.9262 | 1113.0000 | 1991.11.28 | 1992.06.16 | WaterIsotopes.org |
| NIGRO | 40.9667 | 15.6833 | 385.0000 | 2002.04.15 | 2004.03.15 | GNIP |
| Ningqian | 37.3700 | 101.4900 | | 2021.11.27 | 2021.11.27 | Measured data |
| NIROB | 56.7193 | 60.7333 | 150.0000 | 2016.12.15 | 2018.06.15 | GNIP |
| NOGUERA DE ALBARRACIN | 40.4580 | -1.5987 | 1449.0000 | 2013.01.15 | 2015.12.15 | GNIP |
| Nonoichi | 36.3000 | 136.3600 | 40.0000 | 2013.02.01 | 2013.12.29 | WaterIsotopes.org |
| NORDERNEY | 53.7122 | 7.1519 | 11.0000 | 1997.06.15 | 2009.12.15 | GNIP |
| NOVOSIBIRSK | 55.0300 | 82.9000 | 162.0000 | 1990.01.15 | 1990.12.15 | GNIP |
| NRK | 36.1226 | 137.6298 | 1446.0000 | 2010.07.15 | 2015.06.15 | WaterIsotopes.org |
| NSK | 35.8264 | 137.8624 | 1247.0000 | 2010.07.15 | 2013.08.15 | WaterIsotopes.org |
| NY ALESUND | 78.9167 | 11.9333 | 7.0000 | 1990.01.15 | 2016.12.15 | GNIP |
| NYON | 46.3986 | 6.2338 | 436.0000 | 1992.07.15 | 2019.12.15 | GNIP |
| ODENSE | 55.4700 | 10.3300 | 17.0000 | 1963.03.15 | 1984.11.15 | GNIP |
| ODESSA | 46.4800 | 30.6300 | 64.0000 | 1981.02.15 | 1983.12.15 | GNIP |

| | | | | | |
|---|---|---|---|---|---|
| Okayama | 34.4100 | 133.5500 | 7.0000 | 2013.02.02 | 2014.01.06 | WaterIsotopes.org |
| OLENEK | 68.5000 | 112.4333 | 220.0000 | 1996.05.15 | 2000.08.15 | GNIP |
| OMC | 36.5053 | 137.8703 | 776.0000 | 2010.07.15 | 2015.06.15 | WaterIsotopes.org |
| OMSK | 55.0100 | 73.3800 | 94.0000 | 1990.01.15 | 1990.11.15 | GNIP |
| ORLEANS-LA-SOURCE | 47.8332 | 1.9404 | 109.0000 | 1996.03.15 | 2014.12.15 | GNIP |
| PAGANELLA | 46.2000 | 11.0000 | 2125.0000 | 1999.09.30 | 1999.09.30 | WaterIsotopes.org |
| PALLANZA | 45.9167 | 8.5500 | 208.0000 | 2000.12.31 | 2000.12.31 | WaterIsotopes.org |
| PALMA DE MALLORCA | 39.5534 | 2.6253 | 3.0000 | 2000.02.15 | 2015.11.15 | GNIP |
| PALMYRA | 34.5500 | 38.3000 | 400.0000 | 1989.12.15 | 1993.04.15 | GNIP |
| Panzhihua | 26.8900 | 105.0000 | | 2021.12.26 | 2022.05.09 | Measured data |
| PARMA | 44.8000 | 10.3333 | 55.0000 | 2002.02.28 | 2002.02.28 | WaterIsotopes.org |
| PASSO PRESOLANA | 45.5500 | 10.6000 | 1290.0000 | 1999.01.31 | 1999.01.31 | WaterIsotopes.org |
| PATNA | 25.5736 | 85.0703 | 60.0000 | 2003.08.15 | 2005.10.15 | GNIP |
| PATRAS | 38.2800 | 21.7900 | 100.0000 | 2000.10.15 | 2019.12.15 | GNIP |
| PECHORA | 65.1167 | 57.1000 | 56.0000 | 1981.01.15 | 1983.12.15 | GNIP |
| PERM | 58.0100 | 56.1800 | 161.0000 | 1981.01.15 | 1990.07.15 | GNIP |
| PETROPAVLOVSK KAMCAT | 52.9700 | 158.7500 | 24.0000 | 1996.01.31 | 2000.12.31 | WaterIsotopes.org |
| PETROPAVLOVSK-KAMCHATSKIY | 52.9800 | 158.6500 | 24.0000 | 1996.06.15 | 1999.12.15 | GNIP |
| PETZENKIRCHEN | 48.1500 | 15.1500 | 252.0000 | 1966.03.15 | 1971.08.15 | GNIP |
| PIACENZA | 45.0167 | 9.6667 | 61.0000 | 2001.02.28 | 2001.02.28 | WaterIsotopes.org |
| PIAN DELL ELMO | 43.3431 | 13.0611 | 950.0000 | 2000.10.15 | 2002.05.15 | GNIP |
| Pianosa | 42.5859 | 10.0795 | | 2014.11.01 | 2018.12.01 | References |
| PINGLIANG | 35.5333 | 106.7000 | 1570.0000 | 2003.05.15 | 2004.04.15 | GNIP |
| Pingliang | 35.5428 | 106.6847 | | 2021.01.24 | 2022.02.06 | Measured data |
| PIOMBINO | 42.9167 | 10.5333 | 21.0000 | 2000.01.31 | 2000.01.31 | WaterIsotopes.org |
| Piombino | 42.9328 | 10.5111 | | 1997.10.01 | 2000.01.01 | References |
| PISA | 43.7167 | 10.3833 | 5.0000 | 1999.12.31 | 1999.12.31 | WaterIsotopes.org |

| | | | | | | |
|---|---|---|---|---|---|---|
| Pisa | 43.7177 | 10.4222 | | 1992.11.01 | 2004.12.01 | References |
| PISA (CENTRAL) | 43.7100 | 10.4000 | 12.0000 | 1992.11.15 | 2002.02.15 | GNIP |
| PLITVICE | 44.8806 | 15.6189 | 580.0000 | 2004.01.15 | 2005.06.15 | GNIP |
| PODERSDORF | 47.8517 | 16.8431 | 121.0000 | 1965.05.15 | 1968.12.15 | GNIP |
| PONTA DELGADA | 37.7700 | -25.6500 | 175.0000 | 1990.01.15 | 2016.12.15 | GNIP |
| PONTREMOLI | 44.3667 | 9.8833 | 236.0000 | 2002.04.30 | 2002.04.30 | WaterIsotopes.org |
| Pontremoli | 44.3753 | 9.8763 | | 2000.05.01 | 2002.03.01 | References |
| PONTRESINA | 46.4914 | 9.8982 | 1724.0000 | 1994.07.15 | 2019.12.15 | GNIP |
| Port Blair | 11.6600 | 92.7300 | | 2012.05.02 | 2016.12.11 | WaterIsotopes.org |
| PORTO | 41.1333 | -8.6000 | 93.0000 | 1990.01.15 | 2017.06.15 | GNIP |
| PORTOROZ | 45.4667 | 13.6167 | 2.0000 | 2000.10.15 | 2006.12.15 | GNIP |
| POTENZA-TITO SCALO | 40.6333 | 15.8000 | 632.0000 | 1999.08.31 | 1999.08.31 | WaterIsotopes.org |
| PRAGUE | 50.1162 | 14.3929 | 184.0000 | 2012.10.15 | 2018.12.15 | GNIP |
| PRODROMOS | 34.9500 | 32.8300 | 1378.0000 | 1964.03.15 | 1966.12.15 | GNIP |
| PUERTO DE NAVACERRADA | 40.7931 | -4.0106 | 1894.0000 | 2013.01.15 | 2015.12.15 | GNIP |
| PUSCHINO | 54.1800 | 158.0200 | 303.0000 | 1998.04.15 | 2000.03.15 | GNIP |
| Qinghai-Linye | 37.3200 | 101.5200 | | 2021.11.27 | 2021.11.27 | Measured data |
| Qintu Lake | 39.0300 | 103.3600 | | 2020.07.03 | 2021.10.08 | Measured data |
| QIQIHAR | 47.3833 | 123.9167 | 147.0000 | 1988.02.15 | 1992.12.15 | GNIP |
| RAFAH | 31.2833 | 34.2333 | 73.0000 | 2001.01.15 | 2003.03.15 | GNIP |
| RAMATA N1 | 57.9343 | 24.9804 | 49.0000 | 2016.02.15 | 2017.08.15 | GNIP |
| RAMNICU VALCEA | 45.0353 | 24.2842 | 237.0000 | 2012.01.15 | 2016.11.15 | GNIP |
| RAQQA | 35.8972 | 39.3417 | 246.0000 | 1991.12.15 | 1992.05.15 | GNIP |
| REGENSBURG | 49.0422 | 12.1019 | 365.0000 | 1990.01.15 | 2013.12.15 | GNIP |
| REYKJAVIK | 64.1300 | -21.9300 | 14.0000 | 1992.07.15 | 2018.12.15 | GNIP |
| RIARDO | 41.2667 | 14.1500 | 110.0000 | 1998.12.31 | 1998.12.31 | WaterIsotopes.org |
| RIFUGIO GRAFFER | 46.2000 | 10.6000 | 2263.0000 | 1999.09.30 | 1999.09.30 | WaterIsotopes.org |
| RIGA | 56.9360 | 24.0955 | 2.9300 | 2016.04.15 | 2018.12.15 | GNIP |
| RISHIKESH | 30.1122 | 78.3025 | 356.0000 | 2005.07.15 | 2006.09.15 | GNIP |
| RIVO | 46.5189 | 13.0122 | 615.0000 | 1998.03.31 | 1998.03.31 | WaterIsotopes.org |
| Riyadh, Compound | 24.7956 | 46.7588 | 612.0000 | 2009.04.03 | 2012.12.18 | WaterIsotopes.org |

| | | | | | | |
|---|---|---|---|---|---|---|
| Riyadh, no coords | 24.7200 | 46.7200 | 612.0000 | 1975.01.18 | 1979.01.22 | WaterIsotopes.org |
| Riyadh, Office | 24.6795 | 46.7318 | 612.0000 | 2009.03.28 | 2013.03.23 | WaterIsotopes.org |
| RIZE | 41.0244 | 40.5200 | 136.0000 | 2008.07.15 | 2016.12.15 | GNIP |
| RJAZAN | 54.6167 | 39.7167 | 135.0000 | 1981.01.15 | 1983.12.15 | GNIP |
| ROCCAMONFINA | 41.2833 | 13.9833 | 620.0000 | 2000.10.31 | 2000.10.31 | WaterIsotopes.org |
| Rokkasyo | 40.5700 | 141.2100 | 30.0000 | 2013.02.01 | 2013.12.27 | WaterIsotopes.org |
| ROORKEE | 29.8678 | 77.8939 | 274.0000 | 2003.07.15 | 2006.10.15 | GNIP |
| ROSIA MONTANA | 46.3000 | 23.1300 | 0.0000 | 2015.01.15 | 2016.12.15 | GNIP |
| ROSTOV-NA-DONU | 47.2500 | 39.8200 | 77.0000 | 1981.01.15 | 1990.12.15 | GNIP |
| ROVANIEMI | 66.4969 | 25.7552 | 107.0000 | 2003.11.15 | 2017.12.15 | GNIP |
| RYORI | 39.0300 | 141.8100 | 260.0000 | 1980.01.15 | 2006.09.15 | GNIP |
| S. GEMINI | 42.6167 | 12.5500 | 350.0000 | 2001.02.28 | 2001.02.28 | WaterIsotopes.org |
| S. PELLEGRINO IN ALPE | 44.7333 | 9.9167 | 1520.0000 | 1999.12.31 | 1999.12.31 | WaterIsotopes.org |
| S.Anna di Stazzema | 43.9746 | 10.2727 | | 2014.01.01 | 2018.12.01 | References |
| Saalfelden, HZB-Nr. 103929 | 47.4349 | 12.8509 | 795.0000 | 2007.01.15 | 2013.12.15 | WaterIsotopes.org |
| SABAUDIA | 41.3000 | 13.0167 | 17.0000 | 2001.03.31 | 2001.03.31 | WaterIsotopes.org |
| Saga | 33.1500 | 130.1900 | 5.0000 | 2013.02.01 | 2013.12.21 | WaterIsotopes.org |
| SAGAR | 23.8261 | 78.7625 | 551.0000 | 2003.07.15 | 2005.07.15 | GNIP |
| SAINT CATHRENE | 28.6800 | 34.1000 | 1350.0000 | 2001.03.15 | 2001.04.15 | GNIP |
| Sakai | 34.3200 | 135.2700 | 10.0000 | 2013.02.01 | 2014.01.01 | WaterIsotopes.org |
| SALEHARD | 66.5300 | 66.5300 | 16.0000 | 1996.10.31 | 2000.12.31 | WaterIsotopes.org |
| SALEKHARD | 66.5333 | 66.6667 | 16.0000 | 1996.01.15 | 2000.12.15 | GNIP |
| SALLENT DE GALLEGO - LA SARRA | 42.7897 | -0.3290 | 1460.0000 | 2013.01.15 | 2015.12.15 | GNIP |
| SALUGGIA | 45.2333 | 8.0333 | 194.0000 | 2000.10.15 | 2001.10.15 | GNIP |
| SAN PELLEGRINO IN ALPE | 44.1890 | 10.4763 | 1520.0000 | 1993.03.15 | 2002.02.15 | GNIP |

| | | | | | | |
|---|---|---|---|---|---|---|
| San Pellegrino in Alpe | 44.1893 | 10.4805 | | 1993.03.01 | 2002.02.01 | References |
| Sanchalukou | 37.4300 | 101.5500 | | 2021.11.27 | 2021.11.27 | Measured data |
| SANTA CRUZ DE TENERIFE | 28.4634 | -16.2553 | 35.0000 | 2000.02.15 | 2015.12.15 | GNIP |
| SANTA MARIA DI LEUCA | 39.7964 | 18.3686 | 30.0000 | 1999.10.31 | 1999.10.31 | WaterIsotopes.org |
| SANTANDER | 43.4911 | -3.8006 | 52.0000 | 2000.02.15 | 2015.12.15 | GNIP |
| SARATOV | 51.5667 | 46.0333 | 166.0000 | 1981.01.15 | 1984.09.15 | GNIP |
| SARNICO | 45.6667 | 9.9500 | 197.0000 | 1999.01.31 | 1999.01.31 | WaterIsotopes.org |
| SASSARI | 40.7272 | 8.5603 | 225.0000 | 2000.01.31 | 2000.01.31 | WaterIsotopes.org |
| SATKHIRA | 22.7167 | 89.0833 | 10.0000 | 2015.04.15 | 2018.09.15 | GNIP |
| SCERNI | 42.6333 | 14.0333 | 287.0000 | 2001.09.30 | 2001.09.30 | WaterIsotopes.org |
| SCHLESWIG | 54.5275 | 9.5486 | 43.0000 | 1997.06.15 | 2013.12.15 | GNIP |
| Schneealpe, Lurgbauer | 47.7111 | 15.6369 | | 2005.06.04 | 2005.08.11 | WaterIsotopes.org |
| SEEHAUSEN | 52.8911 | 11.7294 | 21.0000 | 1997.06.15 | 2013.12.15 | GNIP |
| Sendai | 38.1600 | 140.5200 | 50.0000 | 2013.02.01 | 2013.12.31 | WaterIsotopes.org |
| Sera | 34.3500 | 133.3000 | 400.0000 | 2013.02.10 | 2013.12.20 | WaterIsotopes.org |
| Seto | 35.1200 | 137.1000 | 304.0000 | 2013.02.01 | 2014.01.06 | WaterIsotopes.org |
| Sevan | 40.5658 | 45.0083 | 1937.0000 | 2014.07.05 | 2015.04.20 | WaterIsotopes.org |
| SGD | 36.5213 | 138.3497 | 1322.0000 | 2010.07.15 | 2015.06.15 | WaterIsotopes.org |
| SGD2 | 36.5317 | 138.3250 | 1249.0000 | 2010.07.15 | 2015.06.15 | WaterIsotopes.org |
| Shangchigou | 37.6300 | 101.8500 | | 2021.11.27 | 2021.11.27 | Measured data |
| SHB | 50.2500 | 106.2100 | 626.0000 | 2002.11.01 | 2003.09.01 | WaterIsotopes.org |
| SHG | 36.7108 | 138.4947 | 1594.0000 | 2010.07.15 | 2015.06.15 | WaterIsotopes.org |
| SHIJIAZHUANG | 38.0333 | 114.4167 | 80.0000 | 1990.01.15 | 2003.11.15 | GNIP |
| SHILLONG | 25.5700 | 91.8800 | 1598.0000 | 1969.01.15 | 1978.10.15 | GNIP |
| Shirahama | 33.4100 | 135.2000 | 10.0000 | 2013.02.01 | 2013.12.12 | WaterIsotopes.org |

| | | | | | | |
|---|---|---|---|---|---|---|
| Shizuoka | 34.5800 | 138.2200 | 30.0000 | 2013.02.01 | 2014.01.03 | WaterIsotopes.org |
| SIENA | 43.3167 | 11.3500 | 322.0000 | 2002.02.28 | 2002.02.28 | WaterIsotopes.org |
| Siena | 43.3140 | 11.3314 | | 1998.01.01 | 2002.02.01 | References |
| Simf | 44.9800 | 34.1500 | | 2009.11.01 | 2011.12.01 | References |
| SINGAPORE | 1.3500 | 103.9000 | 32.0000 | 1972.12.15 | 1975.12.15 | GNIP |
| SINOP | 42.0250 | 35.1583 | 32.0000 | 2008.07.15 | 2016.12.15 | GNIP |
| SION | 46.2203 | 7.3385 | 482.0000 | 1994.07.15 | 2017.11.15 | GNIP |
| SIRACUSA | 37.0667 | 15.3000 | 17.0000 | 2001.01.31 | 2001.01.31 | WaterIsotopes.org |
| Sisian | 39.5083 | 46.0336 | 1580.0000 | 2014.08.22 | 2015.06.27 | WaterIsotopes.org |
| SON LA | 21.3200 | 103.9100 | 600.0000 | 2017.04.15 | 2018.12.15 | GNIP |
| SP-12 | 45.0522 | 33.9753 | | 2012.01.01 | 2015.12.08 | References |
| ST. GALLEN | 47.4269 | 9.4008 | 779.0000 | 2004.01.15 | 2019.12.15 | GNIP |
| ST. PETERSBURG | 59.9667 | 30.3000 | 4.0000 | 1981.01.15 | 1990.03.15 | GNIP |
| St.2 Ploneralm | 46.6969 | 13.0407 | 1625.0000 | 1995.01.01 | 1995.12.01 | WaterIsotopes.org |
| St.3 Buschgrede | 46.6845 | 13.0341 | 1260.0000 | 1995.01.01 | 1995.12.01 | WaterIsotopes.org |
| St.4 Hanser | 46.6736 | 13.0341 | 910.0000 | 1995.01.01 | 1995.10.01 | WaterIsotopes.org |
| St.6 Dellach/Drau | 46.7414 | 13.0833 | 627.0000 | 1995.01.01 | 1995.09.01 | WaterIsotopes.org |
| STARA LESNA | 49.1362 | 20.3065 | 721.0000 | 1991.08.15 | 1992.02.15 | GNIP |
| STUTTGART | 48.8281 | 9.2000 | 314.0000 | 1990.01.15 | 2013.12.15 | GNIP |
| SUWIEDA | 32.7056 | 36.5700 | 1020.0000 | 1989.12.15 | 1993.04.15 | GNIP |
| Suzhou District | 34.4775 | 100.2390 | | 2021.05.18 | 2022.04.30 | Measured data |
| SVYATOHIRS'K | 49.0400 | 37.5770 | 66.0000 | 2015.04.15 | 2016.12.15 | GNIP |
| SWA | 36.0455 | 138.1087 | 760.0000 | 2010.07.15 | 2015.06.15 | WaterIsotopes.org |
| SYLHET | 24.9100 | 91.8453 | 20.0000 | 2009.05.15 | 2018.08.15 | GNIP |
| TAASTRUP | 55.6700 | 12.3000 | 28.0000 | 1965.11.15 | 1971.03.15 | GNIP |
| Taipei(Taiwan) | 25.1638 | 121.4750 | 365.0000 | 1997.06.30 | 1998.12.31 | WaterIsotopes.org |
| Taiwan_ALS | 23.5100 | 120.8100 | 2413.0000 | 2002.05.01 | 2006.11.01 | WaterIsotopes.org |
| Taiwan_BL | 24.2300 | 121.3200 | 2350.0000 | 2005.06.01 | 2008.01.01 | WaterIsotopes.org |
| Taiwan_CY | 23.4800 | 120.4600 | 27.0000 | 2002.05.01 | 2005.09.01 | WaterIsotopes.org |

| | | | | | | |
|---|---|---|---|---|---|---|
| Taiwan_HC | 24.7500 | 121.0000 | 34.0000 | 2003.01.01 | 2004.12.01 | WaterIsotopes.org |
| Taiwan_HL | 23.9700 | 121.6500 | 50.0000 | 2003.07.01 | 2006.05.01 | WaterIsotopes.org |
| Taiwan_HY | 23.3000 | 120.3200 | 69.0000 | 2002.05.01 | 2003.09.01 | WaterIsotopes.org |
| Taiwan_IL | 24.6800 | 121.8000 | 24.0000 | 1993.07.01 | 1994.09.01 | WaterIsotopes.org |
| Taiwan_KH | 22.6300 | 120.2700 | 2.0000 | 2005.12.01 | 2006.09.01 | WaterIsotopes.org |
| Taiwan_KL | 25.1300 | 121.7100 | 27.0000 | 2004.06.01 | 2006.05.01 | WaterIsotopes.org |
| Taiwan_LS | 24.2700 | 121.1600 | 1980.0000 | 2003.06.01 | 2004.09.01 | WaterIsotopes.org |
| Taiwan_MG | 22.6700 | 120.6900 | 740.0000 | 2005.07.01 | 2006.06.01 | WaterIsotopes.org |
| Taiwan_PL | 23.9700 | 120.9800 | 732.0000 | 1999.04.01 | 2002.01.01 | WaterIsotopes.org |
| Taiwan_PT | 22.0300 | 120.6900 | 54.0000 | 2006.01.01 | 2007.11.01 | WaterIsotopes.org |
| Taiwan_TC | 24.1200 | 120.6800 | 34.0000 | 2001.06.01 | 2007.10.01 | WaterIsotopes.org |
| Taiwan_TD | 22.8400 | 121.0900 | 250.0000 | 2004.05.01 | 2007.11.01 | WaterIsotopes.org |
| Taiwan_TP | 24.9800 | 121.5100 | 5.0000 | 2004.03.01 | 2006.12.01 | WaterIsotopes.org |
| Taiwan_TT | 23.9900 | 120.6900 | 75.0000 | 2001.07.01 | 2006.11.01 | WaterIsotopes.org |
| Taiwan_TY | 24.9500 | 121.0200 | 105.0000 | 2004.01.01 | 2006.11.01 | WaterIsotopes.org |
| Taiwan_WL | 24.3500 | 121.3100 | 1800.0000 | 2006.12.01 | 2008.12.01 | WaterIsotopes.org |
| Taiwan_YL | 23.3300 | 121.3100 | 130.0000 | 2005.01.01 | 2007.08.01 | WaterIsotopes.org |
| Taizhou | 30.2674 | 120.1528 | | 2020.12.14 | 2022.03.12 | Measured data |
| Takamatsu | 34.2000 | 134.2000 | 10.0000 | 2013.02.01 | 2013.12.12 | WaterIsotopes.org |
| TAMBOV | 52.7333 | 41.4667 | 139.0000 | 1982.08.15 | 1983.12.15 | GNIP |
| TARQUINIA | 42.2500 | 11.7500 | 133.0000 | 2001.05.31 | 2001.05.31 | WaterIsotopes.org |
| TARSOGNO | 44.4500 | 9.6167 | 768.0000 | 2001.03.31 | 2001.03.31 | WaterIsotopes.org |

| TARTU | 58.2639 | 26.4614 | 70.0000 | 2013.07.15 | 2018.12.15 | GNIP |
|---|---|---|---|---|---|---|
| TASHKENT | 41.2700 | 69.2700 | 428.0000 | 1971.01.15 | 1971.12.15 | GNIP |
| TAWAU - AIRPORT | 4.3170 | 118.1210 | 15.0000 | 2007.07.15 | 2008.12.15 | GNIP |
| TBILISI | 41.7500 | 44.7667 | 427.0000 | 2008.04.15 | 2018.12.15 | GNIP |
| TEHERAN EAST | 35.7400 | 51.5800 | 1350.0000 | 2000.10.15 | 2004.05.15 | GNIP |
| TEHRI | 30.3528 | 78.4833 | 640.0000 | 2004.07.15 | 2006.10.15 | GNIP |
| TERNEJ | 45.0300 | 136.6700 | 68.0000 | 1996.05.31 | 2000.12.31 | WaterIsotopes.org |
| TERNEY | 45.0000 | 136.6000 | 68.0000 | 1996.04.15 | 2000.11.15 | GNIP |
| THESSALONIKI | 40.6700 | 22.9600 | 32.0000 | 2000.10.15 | 2003.08.15 | GNIP |
| THONON-LES-BAINS | 46.3722 | 6.4708 | 385.0000 | 1992.02.15 | 2018.12.15 | GNIP |
| TIANJIN | 39.1000 | 117.1667 | 3.0000 | 1988.04.15 | 2001.12.15 | GNIP |
| Tianshui | 34.5809 | 105.7249 | | 2020.12.23 | 2021.06.29 | Measured data |
| TIRUNELVELI | 8.7278 | 77.7183 | 4.0000 | 2003.07.15 | 2004.10.15 | GNIP |
| TKY | 36.1425 | 137.4223 | 1342.0000 | 2011.06.15 | 2013.12.15 | WaterIsotopes.org |
| TODI | 42.7833 | 12.4000 | 400.0000 | 2001.10.31 | 2001.10.31 | WaterIsotopes.org |
| Tokushima | 34.4000 | 134.2900 | 10.0000 | 2013.02.01 | 2013.12.27 | WaterIsotopes.org |
| TOKYO | 35.6800 | 139.7700 | 4.0000 | 1973.01.15 | 1979.12.15 | GNIP |
| Tongling | 30.9200 | 117.8100 | | 2021.02.18 | 2022.02.17 | Measured data |
| Tongren County | 35.5200 | 102.0180 | | 2021.03.30 | 2021.10.26 | Measured data |
| Tongwei | 35.2110 | 105.2422 | | 2020.07.14 | 2021.09.05 | Measured data |
| TOPOLNIKY | 47.9601 | 17.8620 | 118.0000 | 1992.01.15 | 1992.02.15 | GNIP |
| TORTOSA | 40.8203 | 0.4933 | 50.0000 | 2000.02.15 | 2015.11.15 | GNIP |
| TOULOUSE (UNIV. ECOLAB) | 43.5583 | 1.4694 | 156.0000 | 2011.07.15 | 2012.12.15 | GNIP |
| TRENT | 46.0667 | 11.1333 | 312.0000 | 1997.12.31 | 1997.12.31 | WaterIsotopes.org |
| TRIER | 49.7478 | 6.6581 | 265.0000 | 1990.01.15 | 2013.12.15 | GNIP |
| TRIESTE | 45.6486 | 13.7800 | 14.0000 | 1997.12.31 | 2003.12.15 | WaterIsotopes.org |
| UDH | 47.3200 | 110.6700 | 1033.0000 | 2002.10.01 | 2003.09.01 | WaterIsotopes.org |
| UDINE | 46.0619 | 13.2422 | 113.0000 | 1996.12.31 | 1996.12.31 | WaterIsotopes.org |

| | | | | | | |
|---|---|---|---|---|---|---|
| UHLIRSKA | 50.8325 | 15.1478 | 823.0000 | 2006.05.15 | 2018.12.15 | GNIP |
| UIT_001 | 69.6800 | 18.9600 | 73.0000 | 2019.09.06 | 2021.09.05 | WaterIsotopes.org |
| Uji | 34.5400 | 135.4800 | 25.0000 | 2013.02.01 | 2013.07.31 | WaterIsotopes.org |
| ULAANBAATAR | 47.9333 | 106.9833 | 1338.0000 | 1990.06.15 | 2001.03.15 | GNIP |
| UMN-EL-JEMAL | 32.3055 | 36.2447 | 575.0000 | 1968.11.15 | 1969.04.15 | GNIP |
| URBINO | 43.7167 | 12.6333 | 485.0000 | 2001.10.31 | 2001.10.31 | WaterIsotopes.org |
| Utsunomiya | 36.3300 | 139.5400 | 115.0000 | 2013.02.01 | 2014.01.07 | WaterIsotopes.org |
| UTTARKASHI | 30.7292 | 78.4467 | 1140.0000 | 2004.06.15 | 2006.10.15 | GNIP |
| VAIRANO | 41.3333 | 14.1333 | 155.0000 | 1998.08.31 | 1998.09.30 | WaterIsotopes.org |
| VALENCIA | 39.4806 | -0.3664 | 11.0000 | 2000.03.15 | 2010.12.15 | GNIP |
| VALENTIA | 51.9300 | -10.2500 | 9.0000 | 1990.01.15 | 2018.12.15 | GNIP |
| VALLADOLID | 41.6408 | -4.7544 | 735.0000 | 2000.02.15 | 2015.12.15 | GNIP |
| VALLE AGRICOLA | 41.4333 | 14.2500 | 680.0000 | 1998.08.31 | 1998.08.31 | WaterIsotopes.org |
| Vanadzor | 40.8386 | 44.4361 | 1376.0000 | 2014.07.01 | 2015.04.22 | WaterIsotopes.org |
| VERONA | 45.4500 | 11.0000 | 59.0000 | 1997.12.31 | 1997.12.31 | WaterIsotopes.org |
| VIENNA | 48.2486 | 16.3564 | 198.0000 | 1990.02.15 | 2020.12.15 | GNIP |
| VILLACHER ALPE | 46.6031 | 13.6714 | 2156.0000 | 1990.01.15 | 2002.12.15 | GNIP |
| VILSANDI | 58.3828 | 21.8142 | 6.0000 | 2013.07.15 | 2018.12.15 | GNIP |
| VOLOGDA | 59.2833 | 39.8667 | 118.0000 | 1981.01.15 | 1983.12.15 | GNIP |
| Wadi Hanifa, Riyadh | 24.6200 | 46.6600 | 612.0000 | 1974.03.28 | 1974.03.28 | WaterIsotopes.org |
| WALLINGFORD | 51.6000 | -1.1000 | 48.0000 | 1990.01.15 | 2015.12.15 | GNIP |
| WASSERKUPPE RHOEN | 50.4972 | 9.9428 | 921.0000 | 1990.01.15 | 2013.12.15 | GNIP |
| Weather station Burabay new | 53.1296 | 70.2799 | 309.0000 | 2016.04.24 | 2016.05.08 | WaterIsotopes.org |
| WEIL AM RHEIN | 47.5958 | 7.5944 | 249.0000 | 1990.01.15 | 2013.12.15 | GNIP |
| WELLAMPITIYA | 6.9506 | 79.8795 | 5.0000 | 2009.01.15 | 2018.12.15 | GNIP |
| WIERINGERWERF | 52.8000 | 5.0500 | -4.0000 | 1990.01.15 | 1992.04.15 | GNIP |
| WROCLAW | 51.1165 | 17.0295 | 118.0000 | 2004.05.15 | 2009.05.15 | GNIP |
| WUERZBURG | 49.7703 | 9.9578 | 268.0000 | 1990.01.15 | 2013.12.15 | GNIP |
| Wuhai | 39.6700 | 106.8200 | | 2021.04.24 | 2021.07.16 | Measured data |

| | | | | | | |
|---|---|---|---|---|---|---|
| WUHAN | 30.6200 | 114.1300 | 23.0000 | 1992.06.15 | 1998.05.15 | GNIP |
| WULUMUQI | 43.7800 | 87.6200 | 918.0000 | 1986.02.15 | 2003.12.15 | GNIP |
| Wuqia | 39.7200 | 75.2500 | | 2012.08.01 | 2019.03.01 | References |
| XIAN | 34.3000 | 108.9300 | 397.0000 | 1985.09.15 | 1992.07.15 | GNIP |
| Xi'an | 34.2700 | 108.9500 | | 2020.07.21 | 2020.10.13 | Measured data |
| Xianxi | 28.3143 | 109.7400 | | 2021.11.27 | 2022.05.19 | Measured data |
| Xichang | 27.8952 | 102.2641 | | 2021.04.28 | 2021.08.13 | Measured data |
| Xinghai County | 35.5900 | 99.9900 | | 2021.03.27 | 2021.11.03 | Measured data |
| Xiyingwugou | 37.5300 | 102.1000 | | 2019.10.15 | 2021.11.27 | Measured data |
| Ya'an | 30.3000 | 103.0000 | | 2020.08.06 | 2020.10.14 | Measured data |
| Yabulai weather station (YBL) | 39.3000 | 102.7000 | 1250.0000 | 2013.07.18 | 2016.07.24 | WaterIsotopes.org |
| YAKUTSK | 62.0800 | 129.7500 | 107.0000 | 1996.10.15 | 2000.12.15 | GNIP |
| Yamagata | 38.1200 | 140.1800 | 130.0000 | 2013.01.30 | 2013.12.30 | WaterIsotopes.org |
| Yamaguchi | 34.8000 | 131.2800 | 20.0000 | 2013.01.31 | 2013.12.27 | WaterIsotopes.org |
| Yangkou cave | 29.0300 | 107.1800 | 2140.0000 | 2012.03.01 | 2016.12.01 | WaterIsotopes.org |
| YANGOON | 16.7700 | 96.1700 | 20.0000 | 1961.05.15 | 1963.12.15 | GNIP |
| YANTAI | 37.5300 | 121.4000 | 47.0000 | 1986.03.15 | 1991.04.15 | GNIP |
| YCR | 45.0522 | 33.9753 | | 2010.10.01 | 2010.12.01 | References |
| Yerevan | 40.2003 | 44.5183 | 1134.0000 | 2014.07.06 | 2015.06.17 | WaterIsotopes.org |
| Yichun | 27.8000 | 114.3800 | | 2021.12.26 | 2022.04.30 | Measured data |
| YINCHUAN | 38.4833 | 106.2167 | 1112.0000 | 1988.02.15 | 2000.08.15 | GNIP |
| Yokohama | 35.2800 | 139.3500 | 60.0000 | 2013.02.01 | 2013.12.31 | WaterIsotopes.org |
| Yumen | 34.4775 | 100.2390 | | 2021.06.15 | 2022.02.17 | Measured data |
| ZADAR | 44.1200 | 15.2400 | 5.0000 | 2000.09.15 | 2003.12.15 | GNIP |
| ZAGREB-GRIC | 45.8167 | 15.9833 | 157.0000 | 1996.01.15 | 2003.12.15 | GNIP |
| ZARAGOZA AEROPUERTO | 41.6606 | -1.0042 | 263.0000 | 2000.03.15 | 2015.11.15 | GNIP |
| ZAVIZAN-VELEBIT | 44.8167 | 14.9833 | 1594.0000 | 2000.09.15 | 2003.12.15 | GNIP |

| | | | | | Measured data |
|---|---|---|---|---|---|
| Zeku | 35.0352 | 101.4644 | | 2021.01.20 | 2021.11.15 | Measured data |
| ZHANGYE | 38.9300 | 100.4300 | 1483.0000 | 1990.01.15 | 2003.11.15 | GNIP |
| ZHENGZHOU | 34.7200 | 113.6500 | 110.0000 | 1986.02.15 | 1992.07.15 | GNIP |
| ZINNWALD | 50.7311 | 13.7514 | 877.0000 | 2001.02.15 | 2014.02.15 | GNIP |
| ZITTAU | 50.9000 | 14.8000 | 240.0000 | 2012.11.15 | 2018.12.15 | GNIP |
| ZUNYI | 27.7000 | 106.8800 | 844.0000 | 1986.02.15 | 1992.05.15 | GNIP |

**Lines 36-37: "development trends of climate change" can probably be revised as something like "regional climate change".**

Reply: Based on your suggestion, we have modified the phrase "developing trends in climate change" in lines 36-37 to "regional climate change." This modification more accurately describes the scope and focus of our study.

The changes to the manuscript are as follows:

Severe fluctuations in climatic elements can alter water circulation processes, affect regional climate change, and even change the evolutionary patterns of ecological environments. Among these, stable isotopes in precipitation are an excellent comprehensive tracer, playing an important role in revealing water cycle processes, climate change information, and mechanisms of water resource use in ecosystems (Bowen et al., 2019; Wang et al., 2022).

**Lines 74-76: need citations here, and also in Lines 85 when quoting "comparative",**

Reply: We accept this suggestion, and the revisions to the manuscript are as follows:

The relationship between predicted data from models and actual measured data is "comparative"(Joussaume et al., 1984).

**Line 111: it is unclear how these isotope data starting from 1961 can help study paleoclimate.**

Reply: Understanding past climate, assessing current climate, and foreseeing future climate are crucial aspects of climate research. A more comprehensive dataset of stable isotopes in precipitation forms the foundation for a profound understanding of the "current climate." Modern observations of stable isotopes in precipitation can be utilized to validate and assess the accuracy of paleoclimate models. By comparing the

model-simulated stable isotope ratios with observational data, the model's performance in simulating past climate states and precipitation patterns can be evaluated. This contributes to enhancing the credibility and predictive capability of paleoclimate models. We have elaborated on this in the "Introduction" section and Section "4.3."

The changes to the manuscript are as follows:

**1. Introduction**

The accurate understanding of precipitation stable isotopes' response to modern climate lays the foundation for paleoclimate reconstruction. On the other hand, using general atmospheric circulation models to simulate isotope circulation is a major method for comparing isotope distributions in precipitation under both modern and ancient conditions (Joussaume et al., 1984; Brady et al., 2019). Simultaneously, the comparison between simulated and observed precipitation stable isotopes provides valuable validation for the physical components of atmospheric circulation models (Joussaume et al., 1984; Ruan et al., 2019). In conclusion, the comprehensive data on stable isotopes in precipitation offer more detailed information about the climate and hydrological systems.

Paleoclimate Reconstruction: Well-established precipitation stable isotope observational data are advantageous for validating paleoclimate models under modern conditions. Simultaneously, they contribute to richer comparative data for stable isotopes in precipitation collected in geological archives.

4.3 Drivers of stable isotope variation in precipitation in Eurasia

Physical models with different driving mechanisms can meet various usage needs, including paleoclimate reconstruction. For example, CAM3 simulation outputs precipitation oxygen isotope data (Lin et al., 2024). Machine learning is a novel approach for predicting stable isotopes in precipitation, and European simulation practices indicate that oxygen isotope simulations have shown good results, while simulations for hydrogen isotopes remain challenging (Nelson et al., 2021). In general, uncertainties in both physical models and machine learning need continuous improvement and refinement through real-world data. Additionally, an accurate

understanding of the influencing factors of stable isotopes in precipitation is fundamental for achieving successful predictions through machine learning.

**Lines 180-181: were these statistical tests performed for individual stations or the averages across all stations? Given the spatial variability, it is more appropriate to carry out the analyses for individual stations or at least some smaller regions than the whole Eurasia.**

Reply: In the original manuscript, statistical tests were based on overall data from all sites. Considering the feedback from both reviewers, we believe that limiting the statistical tests to individual sites or smaller regions is reasonable, especially given the spatial variability across the Eurasian continent. We selected the two climatic zones with the most significant differences (tropical and polar zones) and conducted Mann-Kendall (MK) tests on the temporal trends of stable isotopes in precipitation for these two climatic zones (Fig.3). The rationale for this choice is the relatively unclear boundaries between temperate, frigid, and arid zones. Of course, discussions on the stable isotopic characteristics of precipitation in other climatic zones and types will be detailed in the "Results and Discussion" section of the revised manuscript. For instance, Figure 4 analyzes the temporal trends and distribution of data for different climatic zones.

The changes to the manuscript are as follows:

In addition, we selected the two climatic zones with the most significant differences, namely the tropical and polar zones. The reason for this choice is that the boundaries between temperate, frigid, and arid zones are relatively unclear, with subtle changes in trends. Mann-Kendall (MK) tests were conducted on the temporal variations of stable isotopes in precipitation for both climatic zones (Fig.3). For the tropical climate (A), the stable isotopes of precipitation ($\delta^2H$ and $\delta^{18}O$) exhibit multiple non-significant periods of abrupt changes. There is a significant increasing trend from 1971 to 2005, followed by a non-significant decreasing trend since 2009. Overall, the deuterium excess (d-excess) shows a non-significant decreasing trend, but this trend has weakened after 1990. In the polar climate (E), there is a significant increasing trend before 1973, followed by non-significant periods of both increase and decrease after

1975. However, after 2010, a gradually significant increasing trend is observed. Since 1985, the deuterium excess has undergone a non-significant decreasing process, and after 2010, it gradually reaches a significant increasing trend. The uncertainty of the tests is mainly attributed to the spatiotemporal distribution and volume of the data.

[Figure]

**Fig.3** Time series MK test for temperate (C) and cold (D) climates

**Figure 3 is not discussed in the main text.**

Reply: Figure 3 presents the Mann-Kendall (MK) test results for the time series of all sites. Based on your comments on "Lines 180-181," we recognize that this test indeed did not account for spatial differences. Therefore, in the revised manuscript, we have removed Figure 3. Following the Köppen climate classification, we have divided the stable isotopes in precipitation sites into sub-regions based on climate types. We then reconstructed the time series for stable isotopes in precipitation within these sub-regions and conducted trend and changepoint tests on the new time series. Please refer to our response to "Lines 180-181" for details on this modification.

**Lines 207-210: the increase of temporal variance is quite substantial is Figure 4.**

**Can some statistical tests be carried out to verify the possible causes, e.g., the correlation between the data in Figure 4 and the regional temperature/precipitation. Please also refer to my previous comment related to the potential systematic bias when calculating station-averages.**

Reply: As pointed out in your comments under "**Major comments:**" we overlo oked spatial variability in the time series for all sites, and representing unique values on the time series using the arithmetic mean of the sites. We acknowle dge that this has been a major contributor to the increased errors in the time s eries. To address this, we have conducted climate zoning for the Eurasian conti nent and calculated using a weighted average. However, we did not address ou tliers in d-excess, as it is derived from the calculation of hydrogen and oxygen stable isotopes, as shown in Fig. 4.

[Figure]

**Fig.4** The time series variations of $\delta^2H$, $\delta^{18}O$, and d-excess in the Eurasian continent.

**Lines 252-253: what are "atmospheric precipitation lines"? Is this a common terminology, if not, please explain/clarify in the text.**

Reply: This was an oversight in our writing process. We will correct it in the revised manuscript and verify for similar errors throughout the document. The accurate expression is "meteoric water line." The linear correlation line between hydrogen and oxygen isotopic compositions in atmospheric precipitation globally or in specific regions is referred to as the "meteoric water line." Craig (1961) collected samples of natural atmospheric water (including river water, lake water, rainwater, and snow) from various locations worldwide and determined the concentrations of hydrogen and oxygen stable isotopes—D ($^2$H) and $^{18}$O—using a mass spectrometer. The results showed a linear relationship between the $\delta^2$H and $\delta^{18}$O of hydrogen and oxygen-stable isotopes in global precipitation. It is also known as Craig's Meteoric Water Line or the Global Meteoric Water Line of Craig (GMWL of Craig).

The changes to the manuscript are as follows:

4.2 Seasonal changes in meteoric water line and precipitation stable isotopes

The temporal and spatial variations of stable isotopes in precipitation are greatly influenced by meteorological factors, and the changes in precipitation isotopes are consistent with the climatic regions. Therefore, based on the Köppen climate classification, we performed climate zoning for stable isotopes in precipitation sites. We used the least squares method to fit meteoric water line for different climate regions (Fig.6) and considered the seasonal variations of precipitation stable isotopes in different climate regions (Fig.7). The meteoric water line for different climate types indicate relatively small differences in various climate precipitation amounts in tropical climates. The variations in the slope and intercept of the meteoric water line are determined by the combined effects of precipitation and temperature, with convective precipitation weakening the impact of the "temperature effect." Intense convective rainfall and oceanic water vapour transport bring abundant precipitation to tropical regions.

**Lines 286-289: I couldn't follow the discussions here, what does it mean by "are distributed around the global average"? And how does this indicate that the**

**moisture in Eurasia is strongly influenced by oceanic water vapor?**

Reply: The average d-excess value for global precipitation is 10‰, with d-excess typically indicating the degree of imbalance during the evaporation of seawater sources. It generally depends only on the environmental conditions of the water source region, and this value remains constant throughout the entire process from water evaporation into the atmosphere to eventual condensation and rainfall. The westerly and monsoon systems are the primary atmospheric circulation systems over the Eurasian continent, transporting water vapour from the ocean inland and gradually weakening. Therefore, regions approaching this value are likely influenced by maritime water vapour.

The changes to the manuscript are as follows:

Compared to $\delta^2H$ and $\delta^{18}O$, deuterium excess displays a more stable pattern and is distributed around the global average (10‰). The westerly and monsoon systems are the primary atmospheric circulation systems over the Eurasian continent, carrying water vapour from the ocean inland and gradually weakening. This indicates that the humidity in the vast region of Eurasia is strongly influenced by ocean water vapour. Ocean conditions and large-scale atmospheric circulation changes can have profound effects on the climate environment of the Eurasian continent.

**Lines 310-311: the predictors used here are not independent, e.g., evapotranspiration is affected by the parameters like temperature and wind speed. Does this possible interdependency affect the results and conclusions? Some more careful analyses and discussions are needed to assess the possible interdependency between the different predictors (especially when the importance of predictors is assessed in this case).**

Reply: Random Forest regression models can enhance prediction accuracy and robustness by constructing multiple decision trees and averaging their results. Due to its design, Random Forest can effectively handle correlations between features. Here are key points:

Feature Selection: Random Forest randomly selects a subset of features when building each tree, reducing the impact of high correlations between features, as not all

features are used in every model.

Overfitting Reduction: Correlations may lead to model overfitting, but Random Forest mitigates this risk by averaging predictions across multiple decision trees.

Furthermore, Random Forest regression models are commonly used for prediction and classification. In this study, we employed the Random Forest regression model solely to assess the importance of key meteorological variables on the stable isotopes of precipitation. Therefore, multicollinearity between variables is not a severe issue. For Random Forest regression models, a better practice is to directly assess model performance and variable contributions through cross-validation and feature importance evaluation (Vystavna et al., 2021).

However, we have also made improvements to address the issues with the random forest model (Fig. 8). On one hand, we have removed duplicate values generated during the resampling of meteorological variables and represented stable isotope values of precipitation at the station by using weighted averages, thereby mitigating the impact of data errors on the model. Additionally, we introduced cross-validation for the model (Table S3) to assess the robustness of the results. The findings indicate that the random forest model performs well on both the training and testing sets, with superior predictive performance for the target variable $\delta^{18}O$ compared to the target variable $\delta^2H$. This is evident from the smaller root mean square error (RMSE) and mean absolute error (MAE) for $\delta^{18}O$, along with a higher $R^2$. The comparably higher values of RMSE and MAE for $\delta^2H$ can be attributed, on one hand, to the numerical range and volume of $\delta^2H$, and on the other hand, based on existing studies on stable isotope predictions in precipitation, the values fall within a reasonable range (Nelson et al., 2021). Therefore, we can assert that the assessment of the importance of meteorological variables for stable isotope evaluation in precipitation, as utilized in this study, is reasonable and accurate.

**Table S3** Random Forest Model Assessment Indicators

|  | RMSE | MAE |
|---|---|---|
| $\delta^2H$ | 16.87 | 11.02 |
| $\delta^{18}O$ | 2.17 | 1.42 |

[Figure]

**Fig.8** Results of Random Forest Regression Analysis for $\delta^2H$ and $\delta^{18}O$ in Relation to Meteorological Variables.a: Regression results for the training set of $\delta^{18}O$, b: Regression results